# A Second Order Majorant Algorithm for Nonnegative Matrix Factorization

**Mai-Quyen Pham**  *mai-quyen.pham@imt-atlantique.fr*
*IMT Atlantique; UMR CNRS 6285 Lab-STICC*
*CNRS, IRL 2010 CROSSING Adelaide, Australia*

**Jérémy E. Cohen**  *jeremy.cohen@cnrs.fr*
*Univ Lyon, INSA-Lyon, UCBL, UJM-Saint Etienne, CNRS, Inserm, CREATIS UMR 5220, U1206, F-69100 Villeur-banne, France*

**Thierry Chonavel**  *thierry.chonavel@imt-atlantique.fr*
*IMT Atlantique; UMR CNRS 6285 Lab-STICC*

**Reviewed on OpenReview:** *https: // openreview. net/ forum? id= lm16IQmimK*

## Abstract

Nonnegative Matrix Factorization (NMF) is a fundamental tool in unsupervised learning, widely used for tasks such as dimensionality reduction, feature extraction, representation learning, and topic modeling. Many algorithms have been developed for NMF, including the well-known Multiplicative Updates (MU) algorithm, which belongs to a broader class of majorization-minimization techniques. In this work, we introduce a general second-order optimization framework for NMF under both quadratic and $\beta$-divergence loss functions. This approach, called Second-Order Majorant (SOM), constructs a local quadratic majorization of the loss function by majorizing its elementwise nonnegative Hessian matrix. It includes MU as a special case, while enabling faster variants. In particular, we propose mSOM, a new algorithm within this class that leverages a tighter local approximation to accelerate convergence. We provide a convergence analysis, showing linear convergence for individual factor updates and global convergence to a stationary point for the alternating version, AmSOM. Numerical experiments on both synthetic and real datasets demonstrate that AmSOM is a promising algorithm for NMF.

## 1 Introduction

Nonnegative Matrix Factorization (NMF) is the (approximate) decomposition of a matrix into a product of two low-rank nonnegative factor matrices. Beyond dimensionality reduction, NMF provides interpretable decompositions across various domains—from chemometrics and audio source separation to topic modeling, hyperspectral imaging, and representation learning. While Paatero and Tapper could be attributed with the modern formulation of NMF (Paatero & Tapper, 1994) as known in the data sciences, its roots are much older and stem from many fields, primarily chemometrics with the Beer Lambert law (Gillis, 2020).

NMF can be formulated as follows: given a nonnegative matrix $\mathbf{V} \in \mathbb{R}_+^{M \times N}$, find two non-negative (entry-wise) matrices $\mathbf{W} \in \mathbb{R}_+^{R \times M}$ and $\mathbf{H} \in \mathbb{R}_+^{R \times N}$ with fixed $R \leq \min(M, N)$, that satisfy

$$\mathbf{V} \approx \mathbf{W}^\top \mathbf{H}. \tag{1}$$

Integer $R$ is often called the nonnegative rank of the approximation matrix $\mathbf{W}^\top \mathbf{H}$ and is fixed *a priori*, typically, $R \ll \min(M, N)$. This low-rank approximation problem is solved by minimizing a user-defined

loss function $\Psi(\mathbf{W}, \mathbf{H})$:

$$\text{Find } \left(\widehat{\mathbf{W}}, \widehat{\mathbf{H}}\right) \in \underset{(\mathbf{W}, \mathbf{H}) \in \mathbb{R}_+^{R \times M} \times \mathbb{R}_+^{R \times N}}{\operatorname{argmin}} \Psi(\mathbf{W}, \mathbf{H}). \tag{2}$$

Classical instances of $\Psi$ are the squared Frobenius norm and the Kullback-Leibler (KL) divergence. These loss functions are part of the larger class of $\beta$-divergences (Basu et al., 1998), see section 3 for the definition. This family of loss functions plays a central role in NMF across various application domains. For example, the Frobenius loss (i.e., $\beta = 2$) is widely used in image reconstruction and compression (Lee & Seung, 2000), in spectral data analysis (Cichocki et al., 2009), or in representation learning (Hoyer, 2004; Modi et al., 2024). The KL-divergence (i.e., $\beta = 1$) is particularly effective for audio source separation (Févotte et al., 2009) and text mining (Févotte & Idier, 2011). Intermediate values, such as $\beta = 1.5$, are sometimes employed in hyperspectral imaging tasks (Févotte & Dobigeon, 2015) and music information retrieval tasks (Bertin et al., 2009).

There is a large body of literature on how to compute solutions to the NMF problem. Existing iterative algorithms may be classified according to two criteria: the loss that they minimize, and whether they alternatively solve for $\mathbf{W}$ and $\mathbf{H}$, that is, whether they perform for each iteration $k \in \mathbb{N}$

$$\mathbf{H}^{k+1} = \underset{\mathbf{H} \in \mathbb{R}_+^{R \times N}}{\operatorname{argmin}} \Psi(\mathbf{W}^k, \mathbf{H}), \tag{3}$$

$$\mathbf{W}^{k+1} = \underset{\mathbf{W} \in \mathbb{R}_+^{R \times M}}{\operatorname{argmin}} \Psi(\mathbf{W}, \mathbf{H}^{k+1}), \tag{4}$$

or update $\mathbf{W}$ and $\mathbf{H}$ simultaneously. Most alternating algorithms for NMF solve each subproblem equation 3 and equation 4 approximately, aiming to reduce the loss rather than to minimize it exactly. For many loss functions, the subproblems equation 3 and equation 4 are strongly convex, and convergence guarantees can be obtained using the theory of alternating minimization (Tupitsa et al., 2021) and block-coordinate methods (Beck, 2015; Beck & Tetruashvili, 2013). Moreover, alternating algorithms generally perform well in practice. For a quadratic loss such as the squared Frobenius norm, state-of-the-art alternating algorithms include Hierarchical Alternating Least Squares (Cichocki et al., 2007; Gillis & Glineur, 2012). For the more general $\beta$-divergence loss, and KL-divergence in particular, the (Alternating) Multiplicative Updates algorithm proposed by Lee and Seung (Lee & Seung, 1999; 2000), that we will denote by (A)MU in the rest of this manuscript, is still state-of-the-art for many datasets (Hien & Gillis, 2021). Therefore, in what follows, we consider mainly alternating algorithms. Nevertheless, several works have explored a joint optimization strategy for matrix factorization. In particular, a non-convex proximal splitting joint-optimization algorithm was recently proposed that may converge faster than the alternating approach (Rakotomamonjy, 2013; Mukkamala & Ochs, 2019). However, their method is limited to Frobenius loss and primarily relies on Newton-like acceleration. More recently, algorithms for the joint optimization problem using the $\beta$-divergence loss were proposed (Vandecappelle et al., 2020; Marmin et al., 2023; Takahashi et al., 2024).

The AMU algorithm is a popular algorithm for computing NMF. It is straightforward to implement a vanilla AMU, and it can be adapted to minimize additional regularization terms, such as sparsity-inducing $\ell_1$ penalty (Hoyer, 2002; Taslaman & Nilsson, 2012), Cohen & Leplat (2025). There have been several works regarding the convergence and the implementation of AMU (Serizel et al., 2016; Takahashi & Seki, 2016; Zhao & Tan, 2018; Soh & Varvitsiotis, 2021; Leplat et al., 2021), but these works do not modify the core of the algorithm: the design of an approximate diagonal Hessian matrix used to define matrix-like step sizes in gradient descent. A downside of AMU, in particular for the squared Frobenius loss, is that its convergence can be significantly slower than other methods, such that HALS or NeNMF (Gillis, 2020).

## 1.1 Contributions

In this work, we propose a new class of algorithms to compute approximate solutions to the NMF problem with positive factor matrices. This class, which generalizes the AMU algorithm, is built using both a second-order approximation of the loss function and a majorization-minimization strategy. Specifically, we approximate the Hessian matrix of the loss function at a given estimate using a diagonal majorant matrix, where for two symmetric matrices $\mathbf{A}$ and $\mathbf{B}$, we say that matrix $\mathbf{A}$ is a majorant of matrix $\mathbf{B}$ if the difference

$\mathbf{A} - \mathbf{B}$ is positive definite. For any positive vector $\mathbf{u}$ and any elementwise nonnegative symmetric positive definite matrix $\mathbf{B}$, the diagonal matrix Diag $\left(\frac{\mathbf{Bu}}{\mathbf{u}}\right)$ in particular forms a valid majorant of $\mathbf{B}$. Choosing matrix $\mathbf{B}$ as the Hessian matrix of the loss essentially yields a local majorization of the loss. Vector $\mathbf{u}$ can be chosen to make this approximation as tight as possible in a specific, theoretically motivated sense, which we detail in the sequel. The resulting class of algorithms is referred to as **Second-Order Majorization (SOM)**. Our main contributions can be summarized as follows, with distinctions drawn between the squared Frobenius loss and the more general (and more challenging) $\beta$-divergence loss:

**Frobenius loss:** we propose the (Alternating) median SOM ((A)mSOM) algorithm, which provably converges to a stationary point of the NMF problem with positive parameters. For a quadratic loss, mSOM can be seen as a majorization minimization algorithm using forward-backward updates with a preconditioner optimizing a median-based criterion. Other criteria are also explored and discussed. The updates have similar complexity as MU updates but are designed to converge faster, which is confirmed experimentally on simulated and realistic datasets. We also prove linear convergence of the iterates of the mSOM algorithm. In practice, the AmSOM algorithm is competitive with state-of-the-art algorithms for NMF with quadratic loss, such as the Hierarchical Alternating Least Squares (Cichocki et al., 2007; Gillis & Glineur, 2012).

$\beta$**-divergence loss:** the mSOM algorithm is extended to non-quadratic loss functions, using the same idea of quadratic majorization of a local quadratic approximation of the loss. The resulting mSOM algorithm is, however, not a global majorization minimization algorithm for $\beta \in [1, 2)$. We prove nevertheless that the mSOM algorithm for $\beta$-divergence loss converges linearly **locally**, in the **noiseless** case. The algorithm is therefore sensitive to initialization. To help with this issue, we propose a scaling strategy that improves on previously proposed initialization scalings (Hien & Gillis, 2021). We also propose to modify the AmSOM algorithm to guarantee numerically that the majorization condition still holds at each iteration, so that the resulting modified AmSOM is shown to converge globally to a stationary point of the NMF problem under positive constraints on the matrix entries. We confirm experimentally that AmSOM often outperforms AMU and other state-of-the-art algorithms for the Kullback-Leibler divergence loss (obtained with $\beta = 1$) on synthetic experiments, but struggles on realistic dataset. While the proposed mSOM algorithm can in principle apply to other learning problems than NMF, the main challenge in its application is the required nonnegativity of the Hessian matrix. We discuss possible extensions in the conclusion.

This work follows a series of algorithms proposed in the literature that make use of partial second-order information, in particular in the case of $\beta$-divergence loss. Hsieh *et. al.* proposed a coordinate descent second-order algorithm (Hsieh & Dhillon, 2011) that efficiently decreases the loss at each iteration, but it lacks convergence guarantees, and each iteration is costly. Their algorithm was later modified to guarantee that each iteration decreases the loss (Hien & Gillis, 2021). Another related family of algorithms is based on mirror gradient descent. The similarity with our approach is that the loss—or a separable majorant of the loss (Takahashi et al., 2024; 2025)—is upper-bounded by a linear term plus a Bregman divergence (Hien & Gillis, 2021; Takahashi et al., 2024; 2025). However, to the best our knowledge, mirror gradient descent for NMF with $\beta$-divergence loss relies on the Burg entropy, that leads to a loose majorization of the $\beta$-divergence. Mirror descent using Burg entropy, without acceleration techniques, may not outperform other baselines algorithms such as AMU (Takahashi et al., 2024). Rather, it enjoys more relaxed convergence conditions, in particular, Lipschitz continuity of the gradient of the loss is not required. Also relevant to this work are extrapolation techniques for block-coordinate descent methods that provably improve convergence speed at a marginal computational loss (Ang & Gillis, 2019; Hien et al., 2025). These extrapolation techniques can be combined with block majorization minimization algorithms, particularly with our proposed approach. This could be studied more deeply in subsequent works.

## 1.2 Structure of the Manuscript

Section 1 introduces the NMF problem, the motivation for our work, and also specifies the position of our work in the literature. Section 2 details the proposed AmSOM algorithm in the case of a quadratic loss. Section 3 extends the AmSOM algorithm to the case of the $\beta$-divergence loss. Section 4 provides more details about the implementation of the proposed algorithms and a scaling rule useful to improve any positive initialization. Finally, section 5 compares the proposed algorithms with state-of-the-art methods to compute NMF on both synthetic and real datasets. All the proofs are provided in the Appendix.

### 1.3 Notation

Uppercase bold letters denote matrices, and lowercase bold letters denote column vectors. A vector with index $n$ denotes the $n$-th column of the corresponding matrix. For example, $\mathbf{v}_n$ will be the $n$-th column of $\mathbf{V} \in \mathbb{R}^{M \times N}$ with $1 \leq n \leq N$ and $\mathbf{v}_n \in \mathbb{R}^M$. In the same way, $v_{a,b}$ denotes entry $(a, b)$ of matrix $\mathbf{V}$. The symbol $\odot$ denotes the Hadamard (entry-wise) product. The division of two matrices or vectors $\frac{\mathbf{u}}{\mathbf{v}}$ is also meant element-wise. $\mathrm{Diag}(\mathbf{v})$ denotes the diagonal matrix with entries defined from vector $\mathbf{v}$. For $\mathbf{A} \in \mathbb{R}^{M \times N}$, we note $\mathbf{A} \geq \epsilon$ (resp. $\mathbf{A} > \epsilon$) if $\mathbf{A} \in [\epsilon, +\infty)^{M \times N}$ (resp. $\mathbf{A} \in (\epsilon, +\infty)^{M \times N}$). $|\mathbf{A}|$ denotes the matrix formed by taking the absolute value of each entry of $\mathbf{A}$, i.e., $|\mathbf{A}| = (|a_{m,n}|)_{1 \leq m \leq M, 1 \leq n \leq N}$. $\mathbf{A} \succeq 0$ (resp. $\mathbf{A} \succ 0$) means that $\mathbf{A} \in \mathbb{R}^{N \times N}$ is positive semi-definite (resp. positive definite), that is, for all $\mathbf{h} \in \mathbb{R}^N$, $\mathbf{h}^\top \mathbf{A} \mathbf{h} \geq 0$ (resp. $\mathbf{h}^\top \mathbf{A} \mathbf{h} > 0$). $\mathbb{1}_N$ denotes the vector of ones with length $N$ and $\mathbb{1}_{M,N} = \mathbb{1}_M \mathbb{1}_N^\top$. The set $\mathbb{R}_+^R$ denotes the nonnegative orthant of dimension $R$, the set $\mathbb{R}_{++}^R$ the positive orthant, and $\mathbb{R}_\epsilon^R$ is the set of vectors of dimension $R$ with entries larger than $\epsilon$.

## 2 Median Second-Order Majorant: Squared Frobenius Norm

The core idea behind the proposed Second-Order Majorant framework is the local approximation of the loss function by a quadratic upper-bound of the second-order Taylor expansion of the loss. We start by studying the simpler case of a quadratic loss, the squared Frobenius norm, for which the second-order expansion is exact. In this setting, we propose an algorithm to solve each nonnegative least squares problem, mSOM, which converges linearly to the global solution. We also prove the convergence of the alternating algorithm AmSOM as a particular case of the Variable Metric Forward-Backward framework Repetti et al. (2015), (Chouzenoux et al., 2016). Section 3 covers the more general and difficult case of $\beta$-divergence loss.

### 2.1 Derivation of the mSOM Algorithm to Update One Factor Matrix

This section derives the proposed mSOM algorithm for updating factor matrix $\mathbf{H}$ in NMF, with fixed factor matrix $\mathbf{W}$, when the loss is the widely used squared Frobenius norm. Updating matrix $\mathbf{W}$ is done similarly. The loss is separable with respect to each column of $\mathbf{H}$, and we therefore restrict our analysis to the update of one column $\mathbf{h}_n$ of $\mathbf{H}$, specifically the $n$-th column, given the corresponding $n$-th column $\mathbf{v}_n$ of the input data matrix $\mathbf{V}$. For simplicity of notation, we omit the column index $n$ in the vectors and their entries. The full matrix updates for both $\mathbf{W}$ and $\mathbf{H}$ are provided in section 2.3. The loss function for vector $\mathbf{h}$ writes

$$\Psi(\mathbf{h}) = \frac{1}{2} \left\| \mathbf{v} - \mathbf{W}^\top \mathbf{h} \right\|_2^2, \tag{5}$$

where we abusively[1] used the notation $\Psi(\mathbf{h})$ as a shorthand for $\Psi(\mathbf{W}, \mathbf{h})$. Minimizing the loss $\Psi$ under nonnegativity constraints amounts to solving a nonnegative least squares problem (Lawson & Hanson, 1995; Bro & Jong, 1997). A key observation is that if the Hessian matrix $\mathbf{W}\mathbf{W}^\top$ is diagonal, NNLS, which solutions can be otherwise costly to obtain, is solved in closed form by projecting the least squares solution on the nonnegative orthant. An important research direction that we pursue in this work is therefore to compute *nice* diagonal approximations of the Hessian matrix.

The starting point of our methodology is to consider the following class of quadratic functions, tight and tangent at a given point $\mathbf{h}$ to the loss $\Psi$:

$$\xi(\mathbf{h}'; \mathbf{h}, \mathbf{A}(\mathbf{h})) = \Psi(\mathbf{h}) + (\mathbf{h}' - \mathbf{h})^\top \nabla \Psi(\mathbf{h}) + \frac{1}{2} \|\mathbf{h}' - \mathbf{h}\|_{\mathbf{A}(\mathbf{h})}^2, \tag{6}$$

where $\mathbf{A}(\mathbf{h})$ is a diagonal matrix with positive diagonal entries and $\|\mathbf{x}\|_{\mathbf{A}}^2 := \langle \mathbf{x}, \mathbf{A}\mathbf{x} \rangle$ is the squared weighted Euclidean norm. Notice that the function $\xi$ is built as the second-order Taylor expansion of the loss $\Psi$ around $\mathbf{h}$, replacing the Hessian matrix with a simpler diagonal matrix $\mathbf{A}(\mathbf{h})$. Because $\mathbf{A}(\mathbf{h})$ is diagonal and invertible, computing the minimizer of $\xi$ with respect to $\mathbf{h}' \in \mathbb{R}_\epsilon^R$ is straightforward and computationally

---

[1]In fact we use both notations $\Psi(\mathbf{h})$ and $\Psi(\mathbf{W}, \mathbf{H})$ throughout this manuscript, the value of the implicit matrix $\mathbf{W}$ in the notation $\Psi(\mathbf{h})$ should be clear from the context.

inexpensive,

$$\underset{\mathbf{h}' \in \mathbb{R}_\epsilon^R}{\mathrm{argmin}}\ \xi(\mathbf{h}'; \mathbf{h}, \mathbf{A}(\mathbf{h})) = \max\left(\mathbf{h} - \mathbf{A}(\mathbf{h})^{-1}\nabla\Psi(\mathbf{h}), \epsilon\right), \tag{7}$$

where the max operator is applied entry-wise. It is sometimes said that the inverse $\mathbf{A}(\mathbf{h})^{-1} = \frac{1}{\mathbf{A}(\mathbf{h})}$ acts as a matrix step-size for projected gradient descent. A small positive constant $\epsilon$ is used to ensure the parameters are positive, which is important to guarantee the invertibility of the matrix $\mathbf{A}(\mathbf{h})$. More generally, positivity of the parameters in NMF is a widespread technique for proving convergence of alternating algorithms (Takahashi & Seki, 2016), avoiding the zero-locking phenomenon (Gillis, 2020) and ensuring numerical stability in practice.

For the squared Frobenius loss $\Psi$ in Eq. equation 6, the Hessian matrix is simply the Gram matrix $\mathbf{W}\mathbf{W}^\top$, and therefore it can be observed that when the entries in $\mathbf{A}(\mathbf{h})$ are larger than the singular values of the Hessian matrix, *i.e.* when $\mathbf{A}(\mathbf{h}) \succeq \mathbf{W}\mathbf{W}^\top$, function $\xi(.; \mathbf{h}, \mathbf{A}(\mathbf{h}))$ is a global quadratic separable majorant to the loss $\Psi$, tight and tangent at vector $\mathbf{h}$. We are interested in crafting a diagonal majorant matrix $\mathbf{A}(\mathbf{h})$ as close as possible to the true Hessian matrix, with as few computational resources as possible.

There is a wide range of choices available for building such a cheap majorant Hessian matrix $\mathbf{A}(\mathbf{h})$. Notably, projected gradient descent is recovered when matrix $\mathbf{A}$ is chosen as $\frac{1}{L}\mathbf{I}$ with $L$ the Lipchitz constant of loss $\Psi$. In what follows, we propose a new approach to obtain a majorant matrix $\mathbf{A}(\mathbf{h})$. We show that for any nonnegative, symmetric positive definite matrix $\mathbf{B}$, the diagonal matrix $\mathrm{Diag}\left(\frac{\mathbf{B}\mathbf{u}}{\mathbf{u}}\right)$ is a majorant of $\mathbf{B}$. This holds in particular when $\mathbf{B}$ is the Hessian matrix of the loss at a given point, allowing us to build a quadratic majorant $\Psi$.

**Proposition 1** *Let $\mathbf{B} \in \mathbb{R}_+^{R \times R}$ a symmetric matrix, and $\mathbf{u} \in \mathbb{R}_{++}^R$ a vector with positive entries. Then, the symmetric matrix $\left(\mathrm{Diag}\left(\frac{\mathbf{B}\mathbf{u}}{\mathbf{u}}\right) - \mathbf{B}\right)$ is positive semi-definite.*

Given the class of majorant functions $\xi$ with $\mathbf{A}_{\mathbf{u}}(\mathbf{h}) = \mathrm{Diag}\left(\frac{\nabla^2\Psi(\mathbf{h})\mathbf{u}}{\mathbf{u}}\right)$ parameterized by a positive vector $\mathbf{u}$, we ask the following question: how may one choose a particular vector $\mathbf{u}$ such that the obtained majorant $\xi_{\mathbf{u}(\mathbf{h})}$ is as close as possible to the actual loss $\Psi$ around $\mathbf{h}$. Before studying in more details the proposed preferred choice for $\mathbf{u}(\mathbf{h})$, we discuss two other options.
**1) $\mathbf{u}(\mathbf{h}) = \mathbf{h}$:** this choice essentially leads to the original Multiplicative Updates algorithm (Lee & Seung, 2000). The update equation 7 indeed simplifies to

$$\max\left(\mathbf{h} \odot \frac{\mathbf{W}\mathbf{v}}{\mathbf{W}\mathbf{W}^T\mathbf{h}}, \epsilon\right), \tag{8}$$

where the computational bottleneck is the computation of the cross products $\mathbf{W}\mathbf{W}^T$ and $\mathbf{W}\mathbf{V}$.
**2) $\mathbf{u}(\mathbf{h}) = \underset{\mathbf{u}>0}{\mathrm{argmin}}\left\|\frac{\nabla^2\Psi(\mathbf{h})\mathbf{u}}{\mathbf{u}}\right\|_\infty$:** this choice amounts to imposing that the largest singular value of the majorant matrix is as small as possible. One can show that the solution to this minmax problem is any eigenvector associated with the largest singular value of $\nabla^2\Psi(\mathbf{h})$ (see Appendix for more details). Hence, under this criterion, the updates boil down to projected gradient descent with adaptive stepsize $\frac{1}{\mu(\mathbf{h})}$ where $\mu(\mathbf{h})$ is the local Lipschitz constant. In the case of a quadratic loss, this is equivalent to gradient descent with fixed stepsize. In the case of a non-quadratic loss function discussed in section 3, this procedure requires computing the largest eigenvalue of the Hessian matrix at each iteration, which is bound to be costly. Such a local Lipschitz constant computation is reminiscent of the adaptive gradient descent algorithm (Malitsky & Mishchenko, 2020). Because the local Lipschitz constant is not cheap to compute and convergence would require additional checks (Malitsky & Mishchenko, 2020), we did not pursue this direction further.

We propose to choose the vector $\mathbf{u}$ such that the diagonal values of $\mathbf{A}_{\mathbf{u}}(\mathbf{h})$ are as small as possible in median,

$$\mathbf{u}(\mathbf{h}) = \mathbf{u}^* = \underset{\mathbf{u}>0}{\mathrm{argmin}}\left\|\frac{\nabla^2\Psi(\mathbf{h})\mathbf{u}}{\mathbf{u}}\right\|_1. \tag{9}$$

This choice yields several advantages. Firstly, as stated in Proposition 2, this positive vector $\mathbf{u}^*$ can be calculated in closed form, and its computation involves quantities readily available from the computation of

the gradient of the loss. It is also independent of the current iterate $\mathbf{h}$. We name this algorithm median Second-Order Majorant (mSOM), where median accounts for the specific choice to minimize the $\ell_1$ norm of the diagonal entries of the majorant Hessian matrix, and second-order majorant refers to the fact that any matrix $\mathrm{Diag}\left(\frac{\nabla^2 \Psi(\mathbf{h})\mathbf{u}}{\mathbf{u}}\right)$ upper bounds the true Hessian matrix and defines implicitly a second-order approximation of the loss $\Psi(\mathbf{h})$ around $\mathbf{h}$.

**Proposition 2** *Let $\mathbf{B}$ be a symmetric matrix of size $R \times R$, with $\mathbf{B} > 0$. The optimization problem*

$$\underset{\mathbf{u}>0}{\arg\min} \sum_{r=1}^{R} \frac{(\mathbf{Bu})_r}{u_r} \tag{10}$$

*has solutions in the following form:*

$$\mathbf{u}^* = \alpha \mathbb{1}_R, \quad \text{for any } \alpha > 0.$$

Second, minimizing the median criterion in equation 9 is equivalent to choosing, within the diagonal matrices in the class $\mathrm{Diag}\left(\frac{\nabla^2 \Psi(\mathbf{h})\mathbf{u}}{\mathbf{u}}\right)$, the element closest to the Hessian matrix in $\ell_1$ norm. Indeed,

$$\mathbf{u} \in \underset{\mathbf{u}\geq\epsilon}{\arg\min} \left\| \mathrm{Diag}\left(\frac{\mathbf{Bu}}{\mathbf{u}}\right) - \mathbf{B} \right\|_1 = \underset{\mathbf{u}\geq\epsilon}{\arg\min} \sum_{r=1}^{R} \left| \frac{\mathbf{b}_r^\top \mathbf{u}}{u_r} - b_{r,r} \right| = \underset{\mathbf{u}\geq\epsilon}{\arg\min} \sum_{r=1}^{R} \frac{\mathbf{b}_r^\top \mathbf{u}}{u_i} \tag{11}$$

This last equality is due to the fact that both $\mathbf{B}$ and $\mathbf{u}$ are positive, and that

$$\frac{\mathbf{b}_r^\top \mathbf{u}}{u_r} - b_{r,r} = \sum_{j\neq r} \frac{b_{rj} u_j}{u_r} \geq 0.$$

The $\ell_1$ norm induces robustness in the Hessian approximation. The supplementary material contains a discussion on the choice of the $\ell_1$ norm versus the $\ell_2$ norm in equation 11.

Third, we prove this choice leads to linear convergence toward the global solution of the subproblem *w.r.t.* matrix $\mathbf{H}$. This is an improvement over Multiplicative Updates, obtained for the choice $\mathbf{u}(\mathbf{h}) = \mathbf{h}$, for which the convergence rate is unknown to the best of our knowledge. At the $k$-th iterate $\mathbf{h}^{(k)}$, the mSOM update takes the form

$$\mathbf{h}^{(k+1)} = \max\left(\mathbf{h}^{(k)} + \gamma \frac{\mathbf{Wv} - \mathbf{WW}^\top \mathbf{h}^{(k)}}{\mathbf{WW}^\top \mathbb{1}_R}, \epsilon\right). \tag{12}$$

Using $\gamma = 1$ amounts to minimizing the local quadratic approximation $\xi(\mathbf{h}'; \mathbf{h}, \mathbf{A}_{\mathbf{u}^*}(\mathbf{h}))$ exactly, but linear convergence is ensured for any $\gamma \in (0, 2)$. In practice, we observe that choosing $\gamma$ close to 2 leads to faster empirical convergence.

**Theorem 1** *If the stepsize $\gamma$ is chosen in $(0, 2)$ and $\mathbf{W} \geq \epsilon > 0$, the mSOM algorithm with Frobenius loss converges linearly with rate $\mu = \|\mathbf{I} - \gamma \frac{\mathbf{WW}^\top}{\mathbf{WW}^\top \mathbb{1}_R}\|_2 < 1$.*

## 2.2 Complexity of mSOM for the Squared Frobenius Norm

The computational complexity of mSOM to update a vector $\mathbf{h} \in \mathbb{R}_+^R$ for a data vector $\mathbf{v} \in \mathbb{R}^M$ is derived as follows. We assume that the matrix-vector product $\mathbf{W}^\top \mathbf{h}$ has a computational complexity of $\mathcal{O}(MR)$ by counting the number of products, although whether this is accurate for modern computer architecture is debatable. The mSOM algorithm relies on the following expensive computations:

- $\mathbf{Wh}$: the complexity is $\mathcal{O}(MR)$.

- $\mathbf{WW}^\top \mathbf{h}$ and $\mathbf{WW}^\top \mathbb{1}_R$: unless dimension $R$ is large ($R$ in general is reasonably small for NMF problems), one may precompute $\mathbf{WW}^\top$ since this same quantity is used across all mSOM iterations. This costs $\mathcal{O}(MR^2)$. The product with vectors $\mathbf{h}$ or $\mathbb{1}_R$ requires $\mathcal{O}(R^2)$ operations.

Therefore, mSOM for Frobenius loss requires $\mathcal{O}(MR^2)$ products at initialization. For a total of $K_{\max}$ iterations, mSOM requires additionnaly $\mathcal{O}(K_{\max}(R^2 + R))$ products. The total complexity is therefore $\mathcal{O}((M + K_{\max})R^2)$. This is exactly the same asymptotic complexity as the MU algorithm, although mSOM involves an additional subtraction. Therefore, mSOM is bound to be slightly slower than MU.

### 2.3 Solving the NMF Problem for the Squared Frobenius Loss with AmSOM

Computing a solution to the NMF problem requires updating both matrices $\mathbf{H}$ and $\mathbf{W}$. The proposed strategy is to use several iterations of mSOM alternatively for each matrix update. The updates for the resulting AmSOM algorithm in matrix form are the following:

$$\mathbf{W} \text{ update: } \max\left(\mathbf{W} + \gamma \frac{\mathbf{HV}^\top - \mathbf{HH}^\top \mathbf{W}}{\mathbf{HH}^\top \mathbb{1}_R \mathbb{1}_M^\top}, \epsilon\right), \tag{13}$$

$$\mathbf{H} \text{ update: } \max\left(\mathbf{H} + \gamma \frac{\mathbf{WV} - \mathbf{WW}^\top \mathbf{H}}{\mathbf{WW}^\top \mathbb{1}_R \mathbb{1}_N^\top}, \epsilon\right). \tag{14}$$

The AmSOM algorithm is summarized in Algorithm 1. It is crucial to note that in AmSOM, each matrix update is performed several times before the other matrix is updated. AmSOM can be described as an alternating optimization algorithm where mSOM is used to solve approximately each subproblem, see section 4 for more implementation details.

We can formally study the convergence of this algorithm as a particular instance of the Variable Metric Forward Backward algorithm (Chouzenoux et al., 2016), since each update in AmSOM is obtained by Majorization-Minimization (MM). In particular, AmSOM converges to a critical point of the loss function (Chouzenoux et al., 2016).

**Proposition 3 ((Chouzenoux et al., 2016))** *Let $(\mathbf{W}^{(k)})_{k\in\mathbb{N}}$ and $(\mathbf{H}^{(k)})_{k\in\mathbb{N}}$ be sequences generated by the AmSOM Algorithm 1. Assume that $\epsilon > 0$ and that the iterates remain bounded[2]. Then there exists positive constants $(\underline{\nu}, \overline{\nu})$ such that*

$$(\forall\, k \geq 1) \quad \underline{\nu} \leq \mathbf{W}^{(k)}\mathbf{W}^{(k)}{}^\top \mathbb{1}_R \mathbb{1}_N{}^\top \leq \overline{\nu},$$

$$(\forall\, k \geq 1) \quad \underline{\nu} \leq \mathbf{H}^{(k)}\mathbf{H}^{(k)}{}^\top \mathbb{1}_R \mathbb{1}_M{}^T \leq \overline{\nu}.$$

*Further assume that the step-sizes $(\gamma_\mathbf{H}^{k,j})_{k\in\mathbb{N}, 0\leq j\leq J_k-1}$ and $(\gamma_\mathbf{W}^{k,i})_{k\in\mathbb{N}, 0\leq i\leq I_k-1}$ are chosen in the interval $[\underline{\gamma}, \overline{\gamma}]$ where $\underline{\gamma}$ and $\overline{\gamma}$ are any positive constants with $0 < \underline{\gamma} < \overline{\gamma} < 2$.*

*Then, the sequence $(\mathbf{W}^k, \mathbf{H}^k)_{k\in\mathbb{N}}$ converges to a critical point $(\widehat{\mathbf{W}}, \widehat{\mathbf{H}})$ of problem equation 2. Moreover, $(\Psi(\mathbf{W}^k, \mathbf{H}^k))_{k\in\mathbb{N}}$ is a nonincreasing sequence converging to $\Psi(\widehat{\mathbf{W}}, \widehat{\mathbf{H}})$.*

Regarding the actual implementation of the updates, it is essential to precompute the Gram matrices $\mathbf{WW}^\top$ and $\mathbf{HH}^\top$, as well as the cross products with the data $\mathbf{WV}$ and $\mathbf{HV}^\top$, before entering each inner loop, since these cross products are the bottleneck in terms of both memory usage and runtime (Gillis & Glineur, 2012). Moreover, the preconditioners, e.g. matrix $\mathbf{WW}^\top \mathbb{1}_R \mathbb{1}_N^\top$ for the $\mathbf{H}$ update, can be computed by first summing columns of the Grammian $\mathbf{WW}^\top$, effectively computing the marginals vector $\mathbf{z} = \mathbf{WW}^\top \mathbb{1}_R$. Further, many modern programming languages allow for broadcasting element-wise operations; since the outer product $\mathbf{z}\mathbb{1}_N^\top$ simply copies the marginals $\mathbf{z}$, it need not be explicitly computed and one may simply divide the (matrix) gradient by marginals $\mathbf{z}$, avoiding needless memory usage.

## 3 Median Second-Order Majorant: Extension to $\beta$-divergence Loss

While the Frobenius loss is frequently encountered when fitting NMF, it is not always a suitable choice. For instance, NMF is often used to factorize time-frequency matrices in music information retrieval problems (Smaragdis & Brown, 2003), and these data matrices exhibit a large dynamic range. Frobenius loss

---

[2]This condition is technical and mild in practice, as it can be checked after running the algorithm.

---

**Algorithm 1** AmSOM in the case of the Frobenius loss

---

**Input:** $\mathbf{V} \in \mathbb{R}_+^{M \times N}$ and $R \in \mathbb{N} \setminus \{0\}$, $R \leq \min(M, N)$, $\epsilon > 0$, maximum number of outer iterations $K_{\max}$.
**Initialization:** $\mathbf{W}^{(0)} \geq \epsilon$ and $\mathbf{H}^{(0)} \geq \epsilon$. For every $k \in \mathbb{N}$, set the number of inner iterations $J_k \in \mathbb{N}$ and
$\quad I_k \in \mathbb{N}$ and choose the stepsizes $(\gamma_{\mathbf{H}}^{k,j})_{0 \leq j \leq J_k - 1}$ and $(\gamma_{\mathbf{W}}^{k,i})_{0 \leq i \leq I_k - 1}$ be positive sequences in $(0, 2)$.
 1: **for** $k = 0, 1, \ldots, K_{\max} - 1$ **do**
 2: $\quad \mathbf{H}^{(k,0)} = \mathbf{H}^{(k)}, \mathbf{W}^{(k,0)} = \mathbf{W}^{(k)}$
 3: $\quad$ Compute $\mathbf{W}^{(k)}\mathbf{W}^{(k)\top}, \mathbf{W}^{(k)}\mathbf{V}, \mathbf{W}^{(k)}\mathbf{W}^{(k)\top}\mathbb{1}_R$.
 4: $\quad$ **for** $j = 0, \ldots, J_k - 1$ **do**
 5: $\qquad \mathbf{H}^{(k,j+1)}$ updated as in equation 14.
 6: $\quad$ **end for**
 7: $\quad \mathbf{H}^{(k+1)} = \mathbf{H}^{(k,J_k)}$
 8: $\quad$ Compute $\mathbf{H}^{(k+1)}\mathbf{H}^{(k+1)\top}, \mathbf{H}^{(k+1)}\mathbf{V}^\top,$
$\quad \mathbf{H}^{(k+1)}\mathbf{H}^{(k+1)\top}\mathbb{1}_R$.
 9: $\quad$ **for** $i = 0, \ldots, I_k - 1$ **do**
10: $\qquad \mathbf{W}^{(k,i+1)}$ updated as in equation 13.
11: $\quad$ **end for**
12: $\quad \mathbf{W}^{(k+1)} = \mathbf{W}^{(k,I_k)}$
13: **end for**
**Output:** $\mathbf{W} = \mathbf{W}^{(K_{\max})}, \mathbf{H} = \mathbf{H}^{(K_{\max})}$

---

would mostly fit the largest entries in the input data; in contrast, the KL-divergence is a homogeneous function of degree one and therefore promotes data fidelity for the smaller data entries. In what follows, we propose to use the mSOM framework to compute NMF with a $\beta$-divergence loss (Basu et al., 1998), with $\beta \in [1, 2]$. In other words, we choose

$$\Psi(\mathbf{W}, \mathbf{H}) = D_\beta \left( \mathbf{V}, \mathbf{W}^\top \mathbf{H} \right) = \sum_{m=1,n=1}^{M,N} d_\beta(v_{m,n}, \mathbf{w}_m^\top \mathbf{h}_n), \tag{15}$$

where $d_\beta$ is the $\beta$-divergence defined as

$$d_\beta(x, y) = \begin{cases} \frac{x}{y} - \log \frac{x}{y} - 1, & \text{if } \beta = 0 \\ x \log \frac{x}{y} - x + y, & \text{if } \beta = 1 \\ \frac{1}{\beta(\beta-1)} \left( x^\beta + (\beta-1)y^\beta - \beta xy^{\beta-1} \right), & \text{otherwise} \end{cases}$$

Recall that $\beta = 1$ corresponds to the KL-divergence, and $\beta = 2$ to the Frobenius loss.

### 3.1 Issues with the mSOM Algorithm for a Non-Quadratic Loss

An immediate issue with the mSOM algorithm as introduced in section 2 is that for any choice of diagonal approximate Hessian matrix $\mathbf{A}(\mathbf{h})$, the function $\xi(\mathbf{h}'; \mathbf{h}, \mathbf{A}(\mathbf{h}))$ can never be a global majorant to the loss $\Psi$, since a quadratic function is always Lipschitz-continous but the $\beta$-divergence tends to infinity at zero when $1 \leq \beta < 2$.

The idea of the mSOM algorithm for a quadratic loss is to build a global majorant of the loss, as tight as possible inside a given family and for a fixed criterion (median distance with the Hessian matrix spectrum). Global majorization is useful for establishing convergence under a Majorization-Minimization framework, while the optimization of the majorant diagonal Hessian matrix is the key ingredient to the convergence speed of the algorithm. To generalize the mSOM algorithm to non-Lipschitz smooth loss, there are at least two possible solutions:

**1) Change the family of majorizing functions**. This is essentially what can be found in the existing literature. Lee and Seung proposed the AMU algorithm for KL-divergence as a MM algorithm (Lee & Seung, 2000), and this was further formalized and studied in (Févotte & Idier, 2011). The resulting algorithm is still state-of-the-art for many NMF problem instances with KL-divergence loss (Hien & Gillis, 2021). Note

that in the Frobenius case, the multiplicative updates are both the minimizer to a quadratic majorant $\xi_{\mathbf{u}}$ with vector $\mathbf{u}$ set as discussed in 2.1, and the minimizer to a non-quadratic majorant (Févotte & Idier, 2011). This is not the main direction we pursue in this work, but we elaborate on this connection in section 3.4 and section 4.4.

**2) Keep the proposed class of local approximations** $\xi$, which are **not majorants of the loss**. The risk with this approach is to obtain diverging iterates, and our main contribution in the following is to provide guarantees for convergence. While the theory around mSOM in this setup is not yet complete, we show below that in the noiseless setting, local convergence is obtained with a linear convergence rate. The main difference with the mSOM algorithm for a quadratic loss (in the noiseless case) is therefore the sensitivity to initialization. To help with this issue, Proposition 4 in section 4 improves on a known result to scale the initial guesses optimally. Finally, using an additional safeguard operation relying on MU, we discuss how to adapt the convergence proof for Variable-Metric-Forward-Backward (Chouzenoux et al., 2016) to guarantee convergence of the alternating algorithm AmSOM.

### 3.2 The mSOM Algorithm with $\beta$-divergence for the Update of Matrix H

Similarly to the quadratic case, we consider the update of a single column $\mathbf{h}_n$ of matrix $\mathbf{H}$, specifically the $n$-th column. As before, we omit the column index $n$ to simplify the notation. Since the loss is separable across columns, the loss function is written as:

$$\Psi(\mathbf{h}) = D_\beta(\mathbf{v}, \mathbf{W}^\top \mathbf{h}) = \sum_{m=1}^{M} d_\beta(v_m, \mathbf{w}_m^\top \mathbf{h}).$$

When $\beta = 1$, *i.e.* for the KL-divergence loss, this writes

$$\sum_{m=1}^{M} \left( v_m \log\left(\frac{v_m}{\mathbf{w}_m^\top \mathbf{h}}\right) + \mathbf{w}_m^\top \mathbf{h} - v_m \right).$$

One can easily derive the gradient and Hessian matrix of the loss $\Psi$ at vector $\mathbf{h}$,

$$\nabla\Psi(\mathbf{h}) = \sum_{m=1}^{M} \left[ (\mathbf{w}_m^\top \mathbf{h})^{\beta-1} - (\mathbf{w}_m^\top \mathbf{h})^{\beta-2} v_m \right] \mathbf{w}_m,$$

$$\nabla^2\Psi(\mathbf{h}) = \sum_{m=1}^{M} \left[ (\beta-1)(\mathbf{w}_m^\top \mathbf{h})^{\beta-2} - (\beta-2)(\mathbf{w}_m^\top \mathbf{h})^{\beta-3} v_m \right] \mathbf{w}_m \mathbf{w}_m^\top.$$

As for the quadratic case, we build a **quadratic** local approximation $\xi$ around $\mathbf{h}$ of the form

$$\xi(\mathbf{h}'; \mathbf{h}, \mathbf{A}_{\mathbf{u}(\mathbf{h})}(\mathbf{h})) = \Psi(\mathbf{h}) + (\mathbf{h}' - \mathbf{h})^\top \nabla\Psi(\mathbf{h}) + \frac{1}{2}\|\mathbf{h}' - \mathbf{h}\|^2_{\mathbf{A}_{\mathbf{u}(\mathbf{h})}(\mathbf{h})}.$$

with $\mathbf{A}_{\mathbf{u}(\mathbf{h})}(\mathbf{h}) = \mathrm{Diag}\left(\frac{\nabla^2\Psi(\mathbf{h})\mathbf{u}(\mathbf{h})}{\mathbf{u}(\mathbf{h})}\right)$ and $\mathbf{u}(\mathbf{h}) = \arg\min_{\mathbf{u}\geq 0}\|\frac{\nabla^2\Psi(\mathbf{h})\mathbf{u}}{\mathbf{u}}\|_1$, leading to $\mathbf{u}(\mathbf{h}) = \mathbb{1}_R$. Contrary to the quadratic case, the quadratic approximation $\xi$ is not a global majorant of the loss $\Psi$. Using Proposition 2, we find that the majorant Hessian matrix $\mathbf{A}_{\mathbf{u}(\mathbf{h})}(\mathbf{h})$ is computed as

$$\mathbf{A}_{\mathbf{u}(\mathbf{h})}(\mathbf{h}) = \mathrm{Diag}\left( \sum_{m=1}^{M} \left[ (\beta-1)(\mathbf{w}_m^\top \mathbf{h})^{\beta-2} - (\beta-2)(\mathbf{w}_m^\top \mathbf{h})^{\beta-3} v_m \right] \mathbf{w}_m \mathbf{w}_m^\top \mathbb{1}_R \right). \tag{16}$$

For KL-divergence in particular, we can reformulate the Hessian matrix computation in matrix form as

$$\mathrm{Diag}\left( (\mathbf{W} \odot \mathbb{1}_{R,R}\mathbf{W}) \frac{\mathbf{v}}{(\mathbf{W}^\top \mathbf{h})^2} \right).$$

Note that the preconditioner here depends on the current iterate $\mathbf{h}$. Finally, one update of the mSOM algorithm to update $\mathbf{h}$ at a given iteration $k$ writes

$$\mathbf{h}^{(k+1)} = \max\left(\mathbf{h}^{(k)} - \gamma \mathbf{A}_{\mathbf{u}(\mathbf{h})}^{-1}(\mathbf{h}^{(k)})\nabla\Psi(\mathbf{h}^{(k)}), \epsilon\right). \tag{17}$$

In particular for the KL-divergence loss,

$$\mathbf{h}^{(k+1)} = \max\left(\mathbf{h}^{(k)} + \gamma\frac{\mathbf{W}\frac{\mathbf{v}}{\mathbf{W}^\top\mathbf{h}^{(k)}} - \mathbf{W}\mathbb{1}_M}{(\mathbf{W}\odot\mathbb{1}_{R,R}\mathbf{W})\frac{\mathbf{v}}{(\mathbf{W}^\top\mathbf{h}^{(k)})^2}}, \epsilon\right). \tag{18}$$

Even though the mSOM algorithm is not a Majorization Minimization algorithm, for $\beta$-divergence with $1 \leq \beta < 2$ it provably converges when there exists an exact solution $\mathbf{v} = \mathbf{W}^\top\mathbf{h}$. More precisely, the iterates of the mSOM algorithm converge linearly to the global solution $\mathbf{h}^*$ when initialized close to $\mathbf{h}^*$, akin to existing convergence results for second-order methods that mSOM takes inspiration from.

**Theorem 2** *Assume that there exists $\mathbf{h}^*$ such that $\mathbf{v} = \mathbf{W}^\top\mathbf{h}^*$. For $\beta \in [1, 2]$, if there exists a nonnegative constant $\eta < \frac{1}{2}$ such that the initialization $\mathbf{h}^{(0)}$ satisfies for all $m \leq M$*

$$|\mathbf{w}_m^\top\mathbf{h}^{(0)} - \mathbf{w}_m^\top\mathbf{h}^*| + 2\|\mathbf{w}_m\|_2\|\mathbf{h}^{(0)} - \mathbf{h}^*\|_2 \leq \eta\mathbf{w}_m^\top\mathbf{h}^* \tag{19}$$

*then the mSOM algorithm with stepsize $\gamma \in (0, 2 - 2(2-\beta)\frac{\eta}{1-\eta})$ and $\mathbf{W} \geq \epsilon > 0$ converges linearly.*

### 3.3 Remarks on the Local Convergence Criterion.

Theorem 2 requires the initial guess $\mathbf{h}^{(0)}$ to satisfy condition equation 19. It may be unclear to the reader if this condition is difficult to satisfy. We simplify the left-hand side and provide a simpler sufficient condition. Denote $\|\mathbf{h}^{(0)} - \mathbf{h}^*\|_2 = e^0$. Then using Cauchy-Schwarz inequality,

$$|\mathbf{w}_m^\top\mathbf{h}^{(0)} - \mathbf{w}_m^\top\mathbf{h}^*| + 2\|\mathbf{w}_m\|_2\|\mathbf{h}^{(0)} - \mathbf{h}^*\|_2 \leq 3\|\mathbf{w}_m\|_2 e^0$$

for all $m \leq M$, which means that local convergence holds when

$$e^0 \leq \frac{\eta}{3}\min_{m\leq M}\frac{\mathbf{w}_m^\top}{\|\mathbf{w}_m\|_2}\mathbf{h}^*. \tag{20}$$

From this bound, we may infer that the mSOM algorithm with $\beta$-divergence loss is ill-suited for sparse datasets or dataset with sparse factors because linear convergence guarantees would require almost perfect initialization. This observation will be mitigated on synthetic experiments in section 5.

### 3.4 Connection with the Multiplicative Updates for KL-divergence

While the usual formulation of MU as a majorization minimization algorithm does not rely on quadratic majorants, we can recover MU updates with the SOM framework. Indeed, if we suppose that $\mathbf{v} = \mathbf{W}^\top\mathbf{h}^*$ exactly, and if one chooses $\mathbf{u}(\mathbf{h}) = \mathbf{h}$, after some simple derivations, the obtained majorant Hessian matrix $\mathbf{A_h}$ leads to multiplicative updates. For instance, for the KL-divergence and under these hypotheses,

$$\mathbf{A_h}(\mathbf{h}) = \mathrm{Diag}\left(\frac{\mathbf{W}\mathbb{1}_M}{\mathbf{h}}\right), \tag{21}$$

and therefore minimizing the local approximation $\xi$ (setting $\gamma = 1$) yields the following update

$$\mathbf{h} \to \max\left(\mathbf{h}\odot\frac{\mathbf{W}\frac{\mathbf{v}}{\mathbf{W}^\top\mathbf{h}}}{\mathbf{W}\mathbb{1}_M}, \epsilon\right),$$

which is, again, the usual multiplicative updates for KL-divergence. We discuss a heuristic improvement to MU based on this observation in section 4.

### 3.5 Complexity of mSOM for the KL-divergence

The costly operations in mSOM for the KL-divergence are

- $\mathbf{W}^{\top}\mathbf{h}$ that appears in the numerator and the denominator, with complexity $\mathcal{O}(MR)$,

- $\mathbf{W} \odot \mathbb{1}_{R,R}\mathbf{W}$, that costs $\mathcal{O}(MR)$,

- $\mathbf{W}(\frac{\mathbf{v}}{\mathbf{W}^{\top}\mathbf{h}} - \mathbb{1}_R)$ that costs $\mathcal{O}(MR)$,

- $(\mathbf{W} \odot \mathbb{1}_{R,R}\mathbf{W}) \frac{\mathbf{v}}{(\mathbf{W}^{\top}\mathbf{h})^2}$ that also costs $\mathcal{O}(MR)$.

Unlike in the Frobenius case, there is no simple way to precompute costly operations. When running mSOM for $K_{\max}$ iterations, the complexity of mSOM is therefore $\mathcal{O}(K_{\max}MR)$. The mSOM algorithm in the KL-divergence case is therefore typically slower than in the quadratic case when $R \ll K_{\max}$ and $R \ll M$. The MU algorithm for KL-divergence requires only two operations in $\mathcal{O}(MR)$, namely products $\mathbf{W}^{\top}\mathbf{h}$ and $\mathbf{W}\frac{\mathbf{v}}{\mathbf{W}^{\top}\mathbf{h}}$, and is therefore slightly faster per iteration.

### 3.6 Solving the NMF Problem for KL-divergence Loss with AmSOM

In this section, we restricted the presentation of AmSOM to KL-divergence for simplicity since the updates are simpler to write, but AmSOM is derived more generally for $\beta$-divergences with $\beta \in [1,2]$ in the same fashion.

We may use a few iterations of the mSOM algorithm to update each matrix $\mathbf{W}$ and $\mathbf{H}$, until convergence. The resulting AmSOM algorithm for KL-divergence, at iteration $k$, has the following updates in matrix form:

$$\mathbf{W}^{(k+1)} = \max\left(\mathbf{W}^{(k)} + \gamma \frac{\mathbf{H}\frac{\mathbf{V}^{\top}}{\mathbf{H}^{\top}\mathbf{W}^{(k)}} - \mathbf{H}\mathbb{1}_{N,M}}{(\mathbf{H} \odot \mathbb{1}_{R,R}\mathbf{H})\frac{\mathbf{V}^{\top}}{(\mathbf{H}^{\top}\mathbf{W}^{(k)})^2}}, \epsilon\right), \tag{22}$$

$$\mathbf{H}^{(k+1)} = \max\left(\mathbf{H}^{(k)} + \gamma \frac{\mathbf{W}\frac{\mathbf{V}}{\mathbf{W}^{\top}\mathbf{H}^{(k)}} - \mathbf{W}\mathbb{1}_{M,N}}{(\mathbf{W} \odot \mathbb{1}_{R,R}\mathbf{W})\frac{\mathbf{V}}{(\mathbf{W}^{\top}\mathbf{H}^{(k)})^2}}, \epsilon\right), \tag{23}$$

where the operation $\mathbf{W} \odot \mathbb{1}_{R,R}\mathbf{W}$ denotes the (broadcasted) elementwise product of each row of matrix $\mathbf{W}$ with the row vector $\mathbb{1}_R^T\mathbf{W}$. We summarize the AmSOM algorithm for KL-divergence in Algorithm 2.

The main problem with AmSOM is to guarantee the convergence of the iterates. Algorithm mSOM in this context only converges locally, and it is possible to build problem instances where, after one iteration, AmSOM outputs a matrix filled with minimal entries $\epsilon$ when poorly initialized. Moreover, the existing results for the convergence of alternating Variable Metric Forward Backward algorithms rely on the majorization minimization framework (Chouzenoux et al., 2016; Li et al., 2018). Therefore, we propose an adaptation of AmSOM meant for convergence analysis: after each update of the form equation 22 and equation 23, the loss function at the proposed update is evaluated and compared to the approximation $\xi$ at the same position. If the value of the approximation $\xi$ is lower than the loss, we instead use a multiplicative update for which the loss is guaranteed to decrease. With this modification, the AmSOM converges to a stationary point under the VMFB framework. The proof boils down to noticing that the convergence result for VMFB in the work of (Chouzenoux et al., 2016) does not rely on the full majorization of the loss by an auxiliary function, but rather on the majorization of the loss at the minimizer of the auxiliary function. In practice, this safeguard is seldom useful when initialization is performed carefully, and it is not implemented in AmSOM for the experiments shown in section 5. On the other hand, we observed that it avoids a dramatic loss increase in the first iterations when initialization is not carefully chosen and therefore it holds practical interest.

---

**Algorithm 2 AmSOM in the case of KL-divergence**

---

**Input:** $\mathbf{V} \in \mathbb{R}_+^{M \times N}$ and $R \in \mathbb{N} \setminus \{0\}$, $R \leq \min(M, N)$, $\epsilon > 0$, maximum number of outer iterations $K_{\max}$.

**Initialization:** $\mathbf{W}^{(0)} \geq \epsilon$ and $\mathbf{H}^{(0)} \geq \epsilon$. For every $k \in \mathbb{N}$, set the number of inner iterations $J_k \in \mathbb{N}$ and $I_k \in \mathbb{N}$ and choose the stepsizes $(\gamma_\mathbf{H}^{k,j})_{0 \leq j \leq J_k - 1}$ and $(\gamma_\mathbf{W}^{k,i})_{0 \leq i \leq I_k - 1}$ be positive sequences in $(0, 2)$.
  Scale the inputs $\mathbf{W}^{(0)}$ and $\mathbf{H}^{(0)}$ as described in section 4.3.
  **for** $k = 0, 1, \ldots, K_{\max} - 1$ **do**
    $\mathbf{H}^{(k,0)} = \mathbf{H}^{(k)}, \mathbf{W}^{(k,0)} = \mathbf{W}^{(k)}$
    Precompute the marginals $\mathbf{W}^{(k)} \mathbb{1}_M$ and $\mathbb{1}_R^\top \mathbf{W}^{(k)}$ then the product $\mathbf{W}^{(k)} \odot \mathbb{1}_R \mathbb{1}_R^\top \mathbf{W}^{(k)}$.
    **for** $j = 0, \ldots, J_k - 1$ **do**
      $\mathbf{H}^{(k,j+1)}$ updated as in equation 23.
    **end for**
    **if** $\sum_n \xi(\mathbf{h}_n^{(k,j+1)}; \mathbf{h}_n^{(k,j)}, \mathbf{A}_{u(x)}) < \Psi(\mathbf{W}^{(k)}, \mathbf{H}^{(k,J_k)})$, perform a MU step instead (optional).
    $\mathbf{H}^{(k+1)} = \mathbf{H}^{(k,J_k)}$
    Precompute the marginals $\mathbf{H}^{(k+1)} \mathbb{1}_N$ and $\mathbb{1}_R^\top \mathbf{H}^{(k+1)}$ then the product $\mathbf{H}^{(k+1)} \odot \mathbb{1}_R \mathbb{1}_R^\top \mathbf{H}^{(k+1)}$.
    **for** $i = 0, \ldots, I_k - 1$ **do**
      $\mathbf{W}^{(k,i+1)}$ updated as in equation 22.
    **end for**
    **if** $\sum_m \xi(\mathbf{W}_m^{(k,i+1)}; \mathbf{W}_m^{(k,i)}, \mathbf{A}_{u(x)}) < \Psi(\mathbf{W}^{(k,I_k)}, \mathbf{H}^{(k+1)})$, perform a MU step instead (optional).
    $\mathbf{W}^{(k+1)} = \mathbf{W}^{(k,I_k)}$
  **end for**
**Output:** $\mathbf{W} = \mathbf{W}^{(K_{\max})}, \mathbf{H} = \mathbf{H}^{(K_{\max})}$

---

# 4 Implementation Details

## 4.1 How to Stop Inner Iterations

Algorithms 1 and 2 provide a minimal set of instructions outlining the proposed AmSOM algorithm. To minimize the time spent when running the algorithm, a dynamic stopping criterion for the inner iterations is sometimes used (Gillis & Glineur, 2012). However, in this work, we resort to simply fixing the number of inner iterations to a small constant, typically $I_k = J_k = 10$. In our preliminary tests, we observed that this choice strikes a good compromise between convergence speed and runtime. On the theoretical side, the convergence of AmSOM holds regardless of the number of inner iterations.

## 4.2 How to Choose the Stepsize

It is well known that the gradient descent algorithm converges for convex functions with Lipschitz continuous gradients when the step size satisfies $0 < \gamma < 2$ (Boyd & Vandenberghe, 2004). However, it is known that using $\gamma \approx 2$ can lead to faster convergence (Nesterov, 2004; Polyak, 1963). In this work, we propose using a constant stepsize of $\gamma = 1.9$. With a quadratic loss this ensures the convergence of the mSOM algorithm, since $\gamma$ must simply be smaller than two. For this stepsize, the convergence of mSOM with $\beta$-divergence loss is obtained in theory only for good initializations.

## 4.3 Improving Initialization by Columnwise Scaling for $\beta$-divergences

Theorem 2 shows linear convergence for the mSOM algorithm with $\beta$-divergence loss in the noiseless case, but requires a good initialization. We also observed in practice that poor initialization could result in divergent behavior in the early stage of the algorithm. While we do not provide a fool-proof method to initialize the (A)mSOM algorithm to satisfy condition equation 19, we propose a simple scaling rule, which, when combined with a single iteration of AMU applied to randomly initialized matrix $\mathbf{H}$, was empirically sufficient in most scenarios tested in this work to start in the basin of attraction.

**Proposition 4** *For fixed values of* $\mathbf{W} > 0$, $\mathbf{H} > 0$, *the optimization problem*

$$\underset{\mathbf{\Lambda} \in \mathbb{R}_+^{n \times n} \, diagonal}{\mathrm{argmin}} \quad D_\beta(\mathbf{V}, \mathbf{W}^\top \mathbf{H} \mathbf{\Lambda}) \tag{24}$$

*has a closed form solution* $\mathbf{\Lambda}^*$ *defined for all* $n \leq N$ *as*

$$\lambda_{nn}^* = \frac{\sum_{m=1}^M v_{mn} \left(\mathbf{w}_m^\top \mathbf{h}_n\right)^{\beta-1}}{\sum_{m=1}^M \left(\mathbf{w}_m^\top \mathbf{h}_n\right)^\beta} = \frac{\mathbf{v}_n^\top \left(\mathbf{W}^\top \mathbf{h}_n\right)^{\beta-1}}{\mathbb{1}_M^\top \left(\mathbf{W}^\top \mathbf{h}_n\right)^\beta}. \tag{25}$$

An almost identical scaling is discussed in previous works (Hien & Gillis, 2021), but applied uniformly over matrix $\mathbf{H}$ instead of columnwise and only for KL-divergence. Using Proposition 4, one may simply rescale the columns of an initial guess $\mathbf{H}^{(0)}$ as the product $\mathbf{H}^{(0)} \mathbf{\Lambda}^*$. It is tempting to also rescale the rows of an initial guess $\mathbf{W}^{(0)}$ using the same method. However, the columns of matrix $\mathbf{H}$ would need to be scaled again. Optimally scaling both matrices amounts to solving

$$\underset{\mathbf{\Lambda_W} \in \mathbb{R}_+^{M \times M}, \; \mathbf{\Lambda_H} \in \mathbb{R}_+^{N \times N} \; \text{diagonal}}{\mathrm{argmin}} \quad D_\beta(\mathbf{V}, \mathbf{\Lambda_W} \mathbf{W}^\top \mathbf{H} \mathbf{\Lambda_H}), \tag{26}$$

and the algorithm that consists of alternatively performing the scaling of Proposition 4 is a variant of the Sinkhorn algorithm (Sinkhorn & Knopp, 1967). Along similar lines, Chi & Kolda (2012) propose to normalize all columns of the factor matrices before each alternating update. For simplicity, we have not used the Sinkhorn algorithm as a preprocessing step in our experiments.

### 4.4 A SOM Adaptation of MU

For fair comparison with AMU in the experiment section, and in order to see the impact of the median optimal choice of vector $\mathbf{u(h)}$, we also present a modification of MU to enhance its speed within the SOM framework.

We have discussed in section 2 and in section 3.4 that MU can be recovered with the SOM framework by choosing $\mathbf{u(h)} = \mathbf{h}$ to build the majorant (and in the KL-divergence case, we also need to assume that the NMF model is exact). Further, to retrieve the classical MU updates, one needs to choose the stepsize $\gamma$ equal to one. In contrast, the mSOM algorithm can converge for a stepsize up to $\gamma \approx 2$. We therefore propose a variant of the MU algorithm coined MUSOM, where the updates are obtained by choosing $\mathbf{u(h)} = \mathbf{h}$ and $\gamma = 1.9$. For instance, with the Frobenius loss, this yields the following updates:

$$\mathbf{W} \text{ update: } \max\left(\mathbf{W} + \gamma \mathbf{W} \odot \frac{\mathbf{HV}^\top - \mathbf{HH}^\top \mathbf{W}}{\mathbf{HH}^\top \mathbf{W}}, \epsilon\right), \tag{27}$$

$$\mathbf{H} \text{ update: } \max\left(\mathbf{H} + \gamma \mathbf{H} \odot \frac{\mathbf{WV} - \mathbf{WW}^\top \mathbf{H}}{\mathbf{WW}^\top \mathbf{H}}, \epsilon\right). \tag{28}$$

In the experiments section, comparing Alternating MUSOM (AMUSOM), AmSOM, and AMU identifies the respective impact on convergence speed of the stepsize $\gamma > 1$ and of the optimal choice of $\mathbf{u(h)}$. Note that the MUSOM algorithm is a heuristic with no convergence guarantees as soon as $\gamma \neq 1$.

## 5 Experimental Results

In this section, we study the performance of the proposed algorithms against baselines for two specific choices of loss functions, the squared Frobenius norm and the Kullback-Leibler divergence, which are arguably the most commonly encountered loss functions in NMF applications. Both synthetic and realistic datasets are used for the comparison. All the experiments and figures below can be reproduced using the Python code shared on the repository attached to this work[3], and were run on a CPU 11th Gen Intel® Core™ i7-11850H ×16 with 32GB of RAM.

---

[3]https://github.com/cohenjer/MM-nmf

There is a large body of literature on computing solutions to NMF, and while we compare our work with several methods known to perform well in general, a deeper comparison of more NMF algorithms on a more diverse set of applications would be highly valuable. We believe this is beyond the scope of this work and deserves significant, dedicated research effort. We have already started working in that direction using a benchmarking library[4] (Moreau et al, 2022).

## 5.1 Summary of Experimental Results

The experiments conducted in this section show experimentally that AmSOM is a promising algorithm. It competes favorably with respect to the state-of-the-art on simulated datasets. It is noticeably fast for the KL-divergence loss at high SNR. For Frobenius loss, AmSOM is competitive with state-of-the-art algorithms on realistic datasets. For KL-divergence loss, contrary to the synthetic experiments, AmSOM struggles to consistently outperform both MUSOM and MU on realistic datasets, and MUSOM is often competitive with the baselines. This shows that more work is required on crafting Hessian approximations for $\beta$-divergences, and that extrapolation techniques are a promising complementary approach (Hien et al., 2025).

While more experiments are required to understand the strengths and limitations of AmSOM, in particular regarding the impact of data and parameter sparsity, the following tests hint towards a favorable use case for AmSOM when precise results are required on a simple dataset, and when a good initialization is available.

## 5.2 Baseline Algorithms

Standard algorithms to compute NMF typically are not the same for Frobenius and KL losses. In particular, for the Frobenius norm, it is reported in the literature (Gillis, 2020) that accelerated Hierarchical Alternating Least Squares (Gillis & Glineur, 2012) is state-of-the-art, while for KL-divergence, the standard MU algorithm proposed by Lee and Seung is still one of the best-performing methods (Hien & Gillis, 2021).

We compare our implementation of the proposed AmSOM algorithm with the following methods:

- Multiplicative Updates (MU) as defined by (Lee & Seung, 1999). MU is still reported as the state-of-the-art for many problems, in particular when dealing with the KL-divergence loss (Hien & Gillis, 2021). Thus, we use it as a baseline.

- Hierarchical Alternating Least Squares (HALS), which only applies to a quadratic loss. We used the implementation provided in `nn-fac` Marmoret & Cohen (2020).

- Nesterov NMF (NeNMF) (Guan et al., 2012), which is an extension of Nesterov Fast Gradient for computing NMF. We observed that NeNMF provided divergent iterates when used to minimize the KL-divergence, so it is only used with the Frobenius loss in our experiments.

- Vanilla Alternating Projected Gradient Descent (APGD) with an aggressive stepsize $1.9/L$ where $L$ is the Lipschitz constant, also only for the Frobenius loss. GD is unreasonably slow for KL-divergence in our experiments, which is unsurprising since KL-divergence is not Lipschitz continuous at zero.

- ScalarNewton algorithm (Hsieh & Dhillon, 2011; Lin & Boutros, 2020; Hien & Gillis, 2021) (SN) that was proposed for the KL-divergence. Similarly to AmSOM, ScalarNewton utilizes second-order information. ScalarNewton however relies on scalar entry-wise update for each matrix $\mathbf{W}$ and $\mathbf{H}$ to obtain cheap updates. We implemented the simpler version, Cyclyc Coordinate Descent (CCD), which has no convergence guarantees but good empirical performance (Hien & Gillis, 2021).

To summarize, using the convention AXXX to name the alternating variant of algorithm XXX, for the Frobenius loss we evaluate the performance of AMU, AHALS, ANeNMF, and APGD against the proposed AmSOM algorithm and the variant AMUSOM of AMU described in section 4. For KL-divergence, we compare the proposed AmSOM with AMU and its variant AMUSOM as well as ASN CCD. Some of these

---

[4]https://github.com/benchopt/benchmark_nmf

methods have no publicly available implementations in Python. We therefore make our own implementations of all non publicly available methods along with the AmSOM algorithm.

Moreover, in order to further compare the proposed algorithm with the baselines, we also solve the (nonlinear) Nonnegative Least Squares (NLS) problem obtained when factor matrix $\mathbf{W}$ is known and fixed (to the ground-truth if known) discussed in sections 2.1 and 3.2. The optimization problem is then convex and we expect all methods to return the same minimal error.

### 5.3 Synthetic Data Setup

To generate synthetic data matrices with approximately low rank, factor matrices $\mathbf{W}$ and $\mathbf{H}$ are sampled element-wise from i.i.d. uniform distributions on $[0, 1]$. The chosen dimensions $M$, $N$, and the rank $R$ change across the experiments and are reported directly on the figures. The simulation setup is different for the Frobenius and KL-divergence cases.

**Frobenius norm:** The synthetic data is formed as $\mathbf{V} = \mathbf{W}^{\top}\mathbf{H} + \sigma\mathbf{E}$ where $\mathbf{E}$ is a noise matrix sampled element-wise from a Gaussian distribution with unit variance. Given a realization of the data and the noise matrix $\mathbf{E}$, the noise level $\sigma$ is chosen so that the Signal to Noise Ratio (SNR) is fixed to a user-defined value reported on the figures or the captions. Results of an experiment where the columns of matrix $W$ are scaled exponentially are reported in Appendix E, but otherwise, no normalization is performed on the factor matrices or the data.

**Kullback-Leibler-divergence:** The synthetic data $\mathbf{V}$ is formed by sampling the distribution $\mathcal{P}(\alpha\mathbf{W}^{T}\mathbf{H})$ where $\mathcal{P}$ is the Poisson distribution and $\alpha$ corresponds to the signal level. The notion of signal-to-noise ratio is less practical in the case of Poisson noise since the SNR depends on the entries of the matrix product $\mathbf{W}^{T}\mathbf{H}$. To keep the presentation homogeneous, we used the convention $\alpha = \frac{1}{2}10^{SNR/10}$ which amounts to assuming a mean value of $\frac{1}{2}$ for the elements of the product $\mathbf{W}^{T}\mathbf{H}$. Furthermore, in the KL-divergence case, typically dense and sparse datasets lead to different performance for algorithms (Hien & Gillis, 2021). Therefore, we also generate sparse matrices $\mathbf{W}$ and $\mathbf{H}$ by hard thresholding half their (smallest) entries to zero. This simulation setup is coined "sparse" in contrast to the previous "dense" setup. An additional experiment where the columns of matrix $W$ are scaled exponentially is reported in Appendix E. Otherwise, no normalization is performed on the factors or the data.

Initial values for all algorithms are identical for each individual run and are generated randomly in the same way as the true underlying factors. Matrix $\mathbf{H}$ is then optimally scaled columnwise using Proposition 4, and we additionally run a single iteration of AMU in the case of KL-divergence for the NMF problem to prevent poor estimates for AmSOM and AMUSOM in the early stage of the algorithm.

The results are collected using a number $P$ of realizations of the above setup. Unless specified otherwise, we choose $P = 100$. All algorithms run for a fixed number of iterations, reported in the figure captions. For the Frobenius loss, we report loss function values normalized by the product of dimensions $M \times N$, denoted as "n. Loss". The curves shown in the figures describe the median loss at each time point on an interpolated time grid computed individually for each algorithm. Experiments are run using toolbox shootout Cohen (2022) to ensure that all the intermediary results are stored and that the experiments are reproducible.

### 5.4 Comparisons on Synthetic Dataset

We start with the experiments on a synthetic dataset, considering the quadratic and non-quadratic cases separately. Convergence plots show the median error at each time step in full line and 0.25 and 0.75 quartiles in error bands. Additionally, winner profiles inspired from (Dolan & Moré, 2002)[5] show the number of times each method was the fastest to reach a given error threshold over all runs. These winner profiles can help visualize if a method is a clear winner when the variability between experiments is high.

---

[5]Performance plots show the number of times each method reached a percentage of the best error achieved. Such performance plots focus on the iteration at the last error.

### 5.4.1 Synthetic Experiments with the Frobenius Loss

Two set of dimensions are used: $[M, N, R] = [1000, 400, 20]$ and $[M, N, R] = [200, 100, 5]$. We set the SNR to 100dB or 40dB for the NMF problem, and to 100dB or 30dB for the NLS problem.

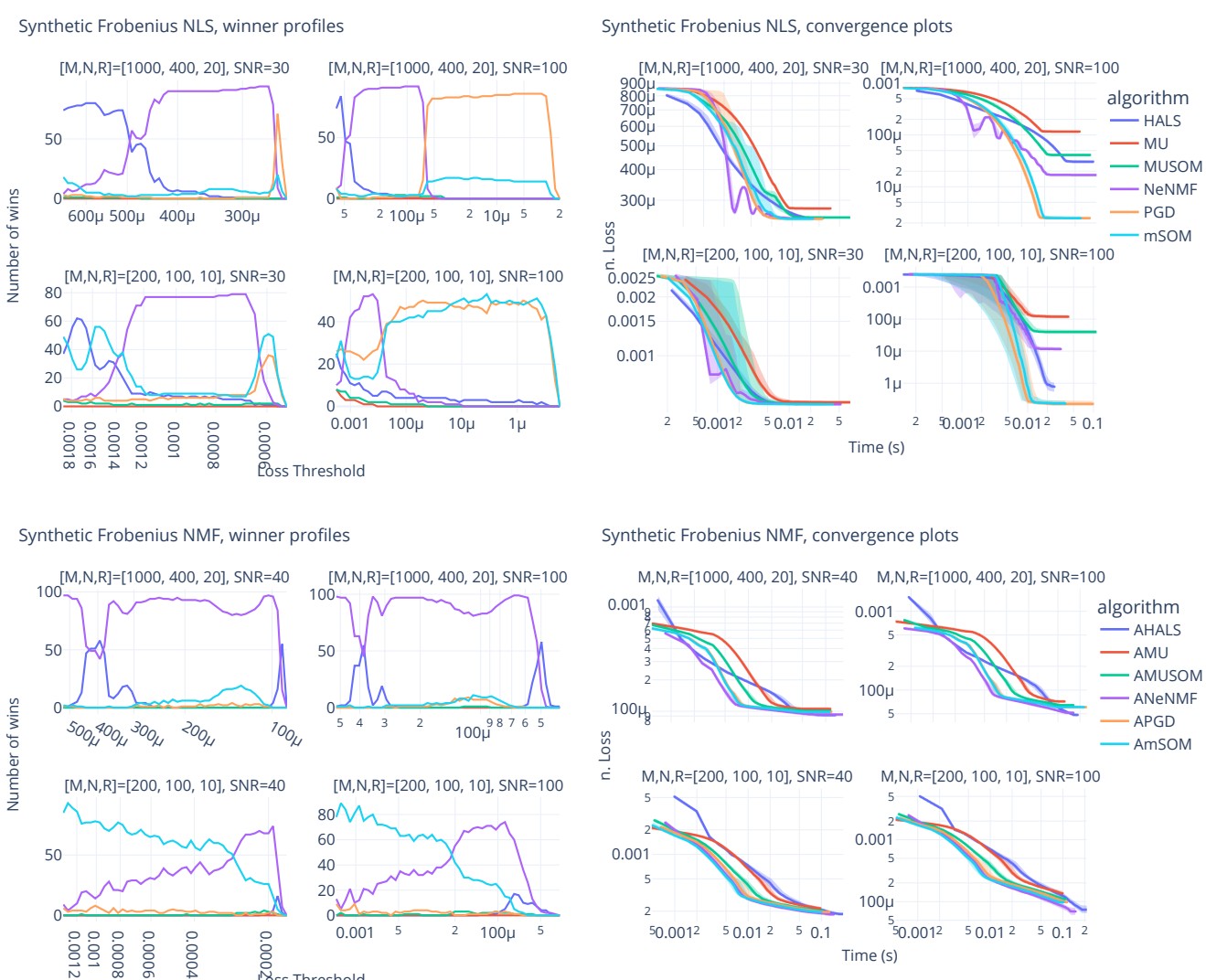

Figure 1: Median plots for the synthetic NLS problem (left) and NMF problem (right) with Frobenius loss with $[M, N, R] = [1000, 400, 20]$ (left) and $[M, N, R] = [200, 100, 5]$ (right), for SNR= 30dB for NLS or 40dB for NMF (top) and SNR $= 100$dB (bottom). The normalized loss function is plotted against time for various methods, and $P = 10$ realizations are computed. The maximal number of iterations is 200 (100 for HALS).

Figure 1 reports the results of the Frobenius loss experiments. We may draw several conclusions from these experiments. First, (A)mSOM has competitive performance with the baselines and, in particular, significantly outperforms AMU. In this experiment, (A)mSOM is also faster than HALS, although this observation will be mitigated in the realistic dataset experiment. Second, the algorithm with extrapolation NeNMF is overall the best performing method for the NMF problem, while PGD and mSOM are performing similarly on the NLS problem. This last observation is due to the columns of $\mathbf{W}$ being close to orthogonality (in which case the Hessian matrix is approximately the identity matrix). Finally, one may observe that the heuristic (A)MUSOM significantly outperforms (A)MU but performs systematically less favorably than

(A)mSOM. This shows that while the use of an aggressive stepsize improves the empirical convergence speed of MU, the choice of a good quadratic majorant is the key element to faster SOM algorithms.

### 5.4.2 Synthetic Experiments with KL-divergence

The dimensions are fixed to $[M, N, R] = [200, 100, 10]$ for both NLS and NMF problems. The SNR is set to either 100 or 20. The number of inner iterations for alternating methods is 10, except for ASN CCD, which performed better with 3 inner iterations.

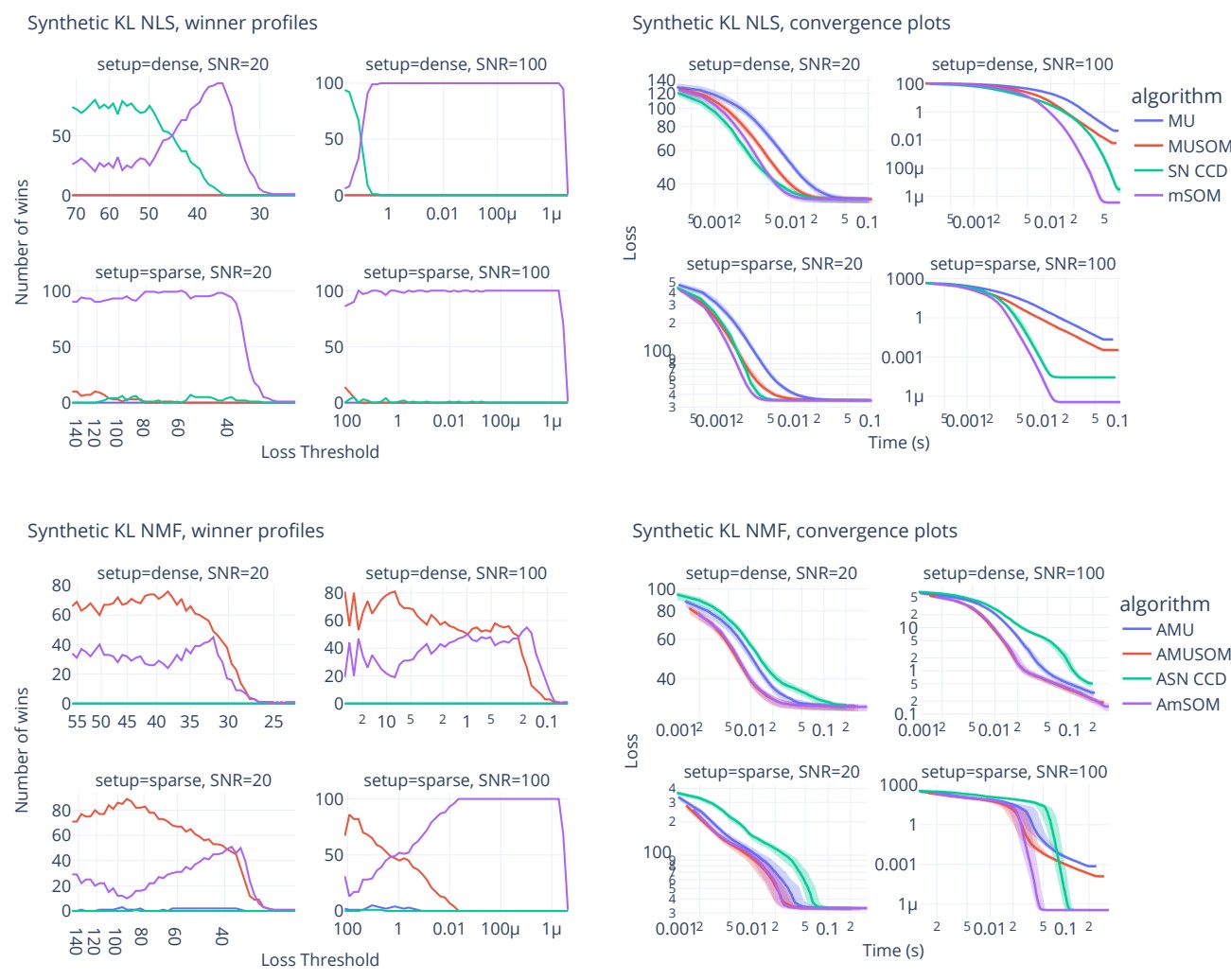

Figure 2: Median plots for the synthetic experiments on the NLS problem (top) and NMF problem (bottom) with the KL-divergence loss. The dimensions are $[M, N, R] = [200, 100, 10]$. The maximal number of iterations is 300 (100 for SN CCD) for NLS, and $K_{\max} = 200$ (40 for ASN CCD) for NMF. Error bands show the 0.25 and 0.75 quartile errors.

Figure 2 reports the KL-divergence loss of all compared methods against time for both the NLS and NMF problems in each setup. The (A)mSOM algorithm converges faster than the baselines for the NLS and NMF problems and in both dense and sparse setups. The iterations of ASN CCD are typically more efficient but have higher time complexity. The time complexity of AMU, AMUSOM, and AmSOM is similar, therefore, the difference in convergence speed can be attributed mainly to the design of a tighter quadratic majorant and the choice of an aggressive stepsize. mSOM is notably faster than MUSOM in the NLS problem, but this does not translate into the NMF problem, where both methods are essentially equivalent except in the

sparse, noiseless regime, where AmSOM significantly outperforms AMUSOM near the optimum. We can infer that mSOM is a promising algorithm to perform NLS with KL-divergence. The AmSOM algorithm is fast for simple problems when initialized close to the solution, or at high SNR. Figure 7 in Appendix E shows further that when the problem is harder because of unbalanced components, AmSOM is the best method at high precision, but does not converge faster than the baselines in early iterations. These results are in line with the convergence theory of AmSOM for *beta*-divergences, which shows linear convergence only near the solution in the noiseless case.

### 5.5 Comparisons on Realistic Datasets

We used the following datasets to further the comparisons:

- A hyperspectral image of dimensions $N = 307^2$ and $M = 162$ called *Urban* which has been used extensively in the blind spectral unmixing literature for showcasing the efficiency of various NMF-based methods, see for instance the survey (Bioucas-Dias et al, 2012). Blind spectral unmixing consists in recovering the spectra of the materials present in the image as well as their spatial relative concentrations. These quantities are in principle estimated as rank-one components in the NMF. It is reported that Urban has between four and six components. We therefore set $R = 6$. These HSI data are dense and moderately-sized.

- An amplitude spectrogram of dimensions $M = 1000$ and $N = 1450$ computed from an audio excerpt in the MAPS dataset (Emiya et al., 2010). More precisely, we hand-picked the file `MAPS_MUS-bach_846_AkPnBcht` which is a recorded recreation of Bach's first Prelude from pure MIDI played by a Yamaha Disklavier. Only the first thirty seconds of the recording are kept, and we also discard all frequencies above 5300Hz. This piece is a simple piano piece with few simultaneous note activations, and NMF can in that case be used efficiently on the amplitude spectrogram to perform blind automatic music transcription, see (Benetos et al., 2018) for more details. The rank of the factorization is a hyperparameter in the set $\{2, 11, 23, 45\}$ corresponding respectively to two notes and one, two or three octaves in the middle of the piano keyboard where the piece is played. KL-divergence is usually preferred over Frobenius loss for measuring discrepancies in spectrograms (Bertin et al., 2009), but both metrics have been used in this context.

In what follows, we are not interested in the interpretability of the results since the usefulness of NMF for music automatic transcription and spectral unmixing has already been established in previous works, and we do not propose any novelty in the model. Instead, we investigate if (A)mSOM and (A)MUSOM variants are faster than the baseline to minimize the loss function on this dataset.

For both datasets, a reasonable ground-truth for the **W** factor is available to compare algorithms for the NLS problem as well. Indeed in the audio transcription problem, we used attack spectra obtained from a model called Attack-Decay (Cheng et al., 2016) trained on the MAPS single notes dataset Wu et al. (2022), in which each rank-one component corresponds to either the attack or the decay of a different key on the piano. In the blind spectral unmixing problem, we use a *fake* ground-truth (Zhu, 2017) where six pixels containing the unmixed spectra have been carefully hand-chosen. For all experiments, the optimal scaling discussed in section 4 is used on columns of matrix **H**.

Figure 3, 4 and 5 respectively show the results for the HSI experiment and the audio NLS and NMF experiments. Since a reasonable ground truth is available for these problems, for the NMF problems we initialized matrix **W** as this ground-truth plus $0.1\mathbf{N}$ for an i.i.d. uniform noise matrix **N** to stabilize the runs. The results complement the observations made in the synthetic experiments. Convergence plots show the minimal error at each time step in full line and 0.5 quartiles in error bands, essentially showing the range of loss values between the minimum and the median. We chose this visualisation since "running the experiment ten times and picking the best run" is a typical procedure for computing NMF in practice.

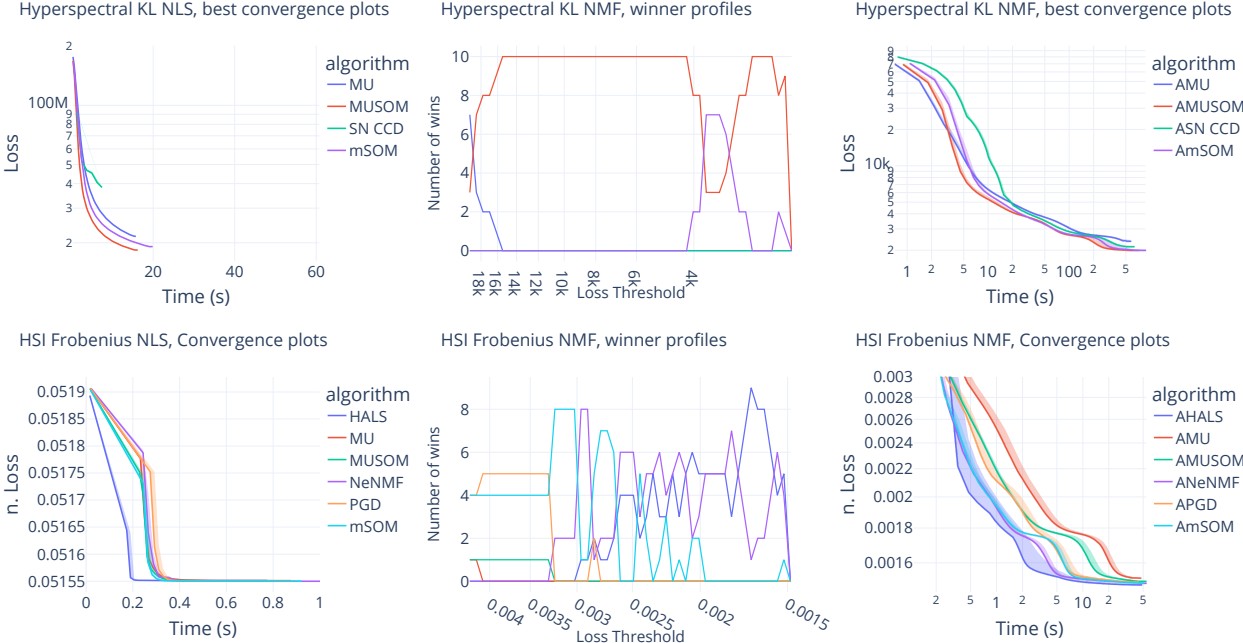

Figure 3: Results of the spectral unmixing experiment, with KL-divergence (top) and Frobenius loss (bottom). Left: NLS experiments with $K_{\max} = 50$ iterations and $P = 100$ runs. Middle and right: NMF experiments with $K_{\max} = 200$ iterations and $P = 10$ runs. The error bands show the range of loss function values from minimal to median values.

### 5.5.1 Results for Spectral Unmixing

**Frobenius loss:** For both NLS and NMF problems, HALS is the best performing method, as is generally reported in the literature. For the NLS problem, all other methods perform overall similarly, with the computation of the Gram matrices in the first iteration being the bottleneck. On the NMF problem, AMU performs poorly. AMUSOM is faster than AMU while still being rather slow. NeNMF, AmSOM, and APGD all behave similarly and are close to the state-of-the-art. In particular, AmSOM is faster than both AMU and AMUSOM. This shows that the curvature obtained by the mSOM procedure may accelerate convergence.

**KL-divergence loss:** Results for the NLS and NMF problems are similar. (A)MUSOM is the fastest algorithm. For the NMF problem, the performance gap with AmSOM vanishes towards convergence, while AMU and ASN CCD struggle to reach low loss function values. These results show that in this context, the larger stepsize $\gamma = 1.9$ used by both AmSOM and MUSOM is the main ingredient, in this experiment, for improving convergence speed.

### 5.5.2 Results for Automatic Transcription

**Frobenius loss:** Overall, all algorithms have comparable performance on the audio dataset with the squared Frobenius norm as a loss function. The NLS plots in Figure 4 show an advantage of mSOM and MUSOM over other methods for small rank values, but as the dimensionality increases, HALS becomes the fastest method. For the NMF problem, while AmSOM is often the fastest method, there is no clear winner. It is interesting to note that AMU performs well on this dataset, even when compared to HALS, which is known to be significantly faster than AMU in most applications.

**KL-divergence loss:** At all rank values, (A)mSOM performs worse than (A)MU and (A)MUSOM in the NMF problem, see Figure 5, but struggles only for larger ranks in the NLS problem, see Figure 4. In the NMF problem, for the rank two case, the loss even increases. This can happen despite our efforts to initialize carefully the factor matrices. The two components selected for initialization and the NLS dictionary in the

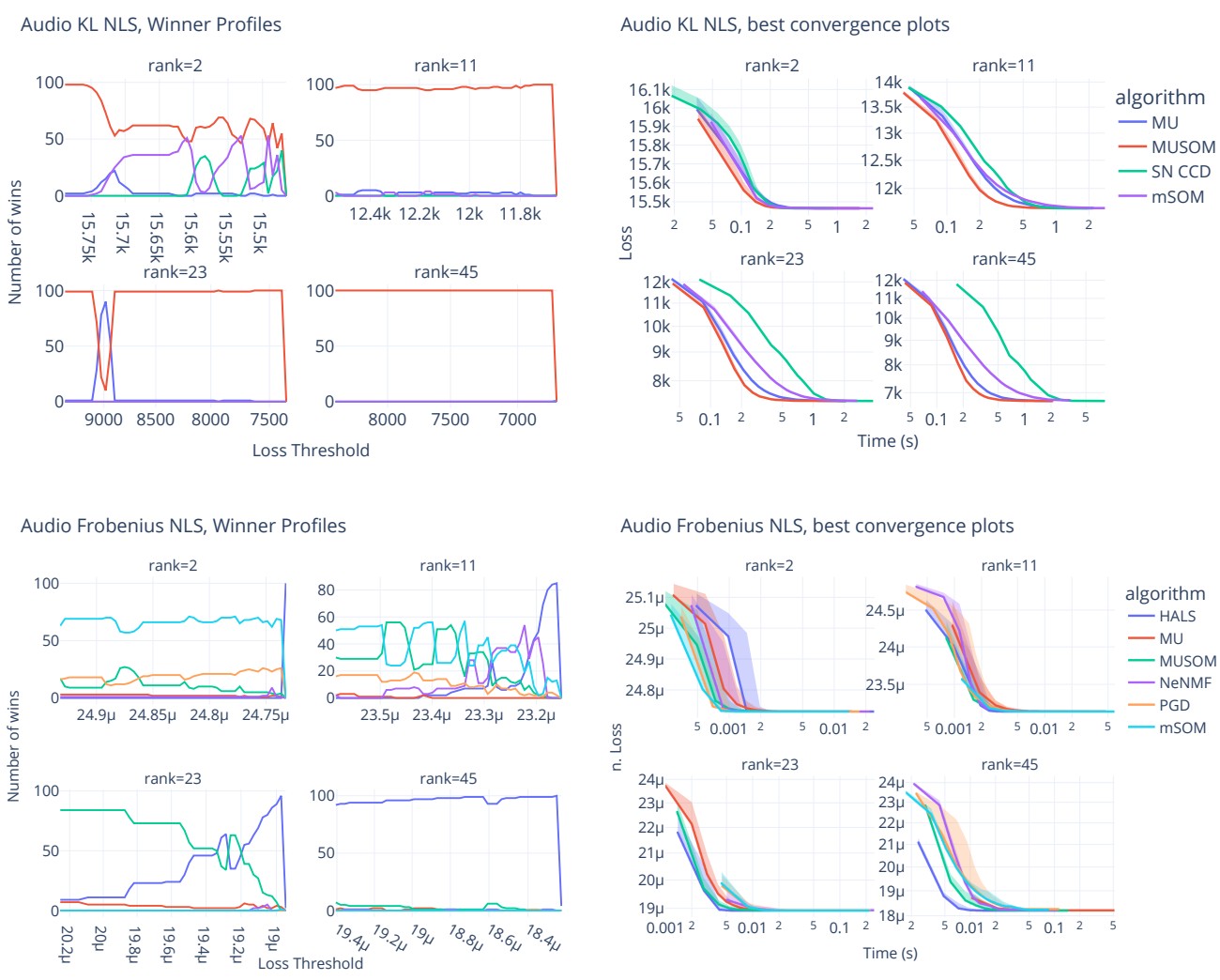

Figure 4: Results of the NLS audio experiment with KL-divergence loss (top) and Frobenius loss (bottom) with $P = 100$ runs. The maximal number of iterations for the NLS and NMF problems is $K_{\max} = 50$ (10 for SN CCD). Error bands show all the runs up to the median loss values.

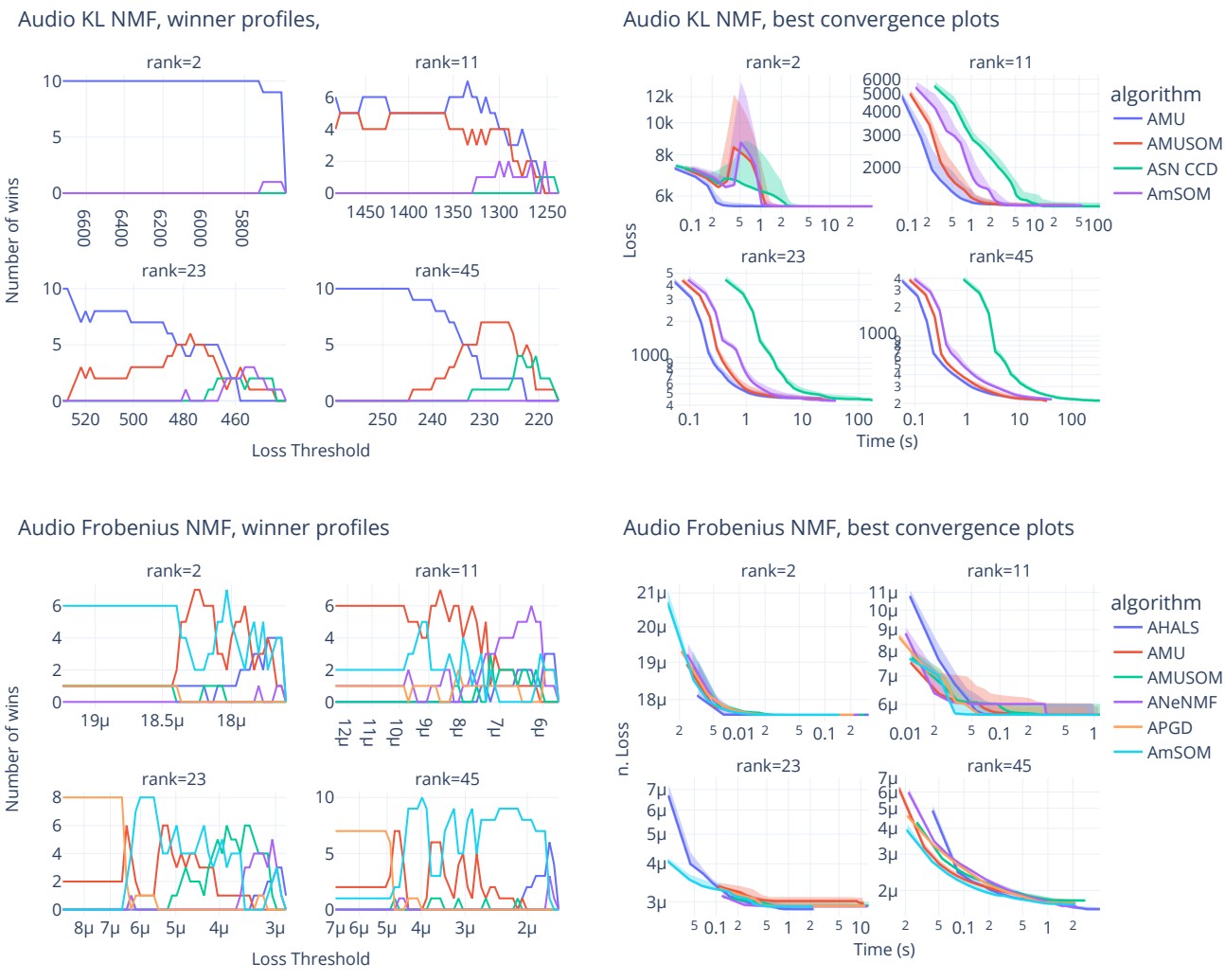

Figure 5: Results of the NMF audio experiment with KL-divergence loss with $P = 10$ runs. The maximal number of iterations for the NMF problems is $K_{\max} = 400$ (100 for SN CCD). Error bands show all the runs up to the median loss values.

rank-two case are not enough to properly model the data. Therefore, the residuals are large, and the local linear convergence from Proposition 2 most likely does not hold. This can explain why the cost increases (note that it also increases for ASN CCD). The underwhelming performance of AmSOM may also be due to the fact that audio data are particularly sparse, and as the rank increases, some notes may not activate at all. As discussed in section 3, mSOM may struggle in the initial phase of the algorithm when the solutions $\mathbf{W}$ and $\mathbf{H}$ are sparse, although this phenomenon was not observed in the synthetic dataset. In the NMF plots in particular, we may observe that after many iterations, AMUSOM and sometimes AmSOM and SN CCD reach a lower loss value than AMU while they struggle to outperform AMU initially. ASN CCD is overall too slow per iteration, in particular when the rank increases. Overall, (A)MUSOM performs better than (A)MU near the solution, but has no convergence guarantees and requires a careful initialization.

## 6 Conclusion and Perspectives

In this work, we propose a tight upper bound of the Hessian matrix encountered in the Nonnegative Matrix Factorization (NMF) problem to improve the convergence speed of the Multiplicative Updates (MU) algo-

rithm for Frobenius and Kullback-Leibler losses. The proposed algorithm, coined AmSOM, and a related extension of MU, coined AMUSOM, show promising performance on both synthetic and real-world data. While in the experiments we conducted, AmSOM is not always better than MU proposed by Lee and Seung, in many cases, it is significantly faster and theoretically motivated. Moreover, for the Frobenius loss, it is competitive with HALS, which is one of the state-of-the-art methods for NMF. The proposed AmSOM algorithm is overall particularly fast in low-noise problems and when properly initialized.

There are many promising research avenues for AmSOM. First, while mSOM requires the Hessian matrix to be elementwise nonnegative, it could be applied in a more general setting by considering a splitting of negative and positive parts of the Hessian (Sha et al., 2007), or its absolute value. Second, this work shows how to build a family of majorant functions that allows for a variety of update rules, including the MU, and how to compute a good majorant in that family. While we choose the best majorant according to a median curvature criterion, other criteria that may give birth to faster algorithms could be explored. We anticipate moreover that an efficient algorithmic procedure to compute a better majorant could be designed. Finally, the developments in joint optimization methods for NMF motivate us to extend the SOM framework by adopting a joint optimization strategy to accelerate convergence.

### Acknowledgments

This research is funded by ANR JCJC LoRAiA project (ANR-20-CE23-0010). The authors are grateful to Nicolas Gillis and Jean-Philippe Diguet for their insightful comments and suggestions on an earlier version of this manuscript.

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

# A   Appendix

## A.1   Proof of Proposition 1 (Hessian majorization)

It is straightforward to prove that a square matrix $\mathbf{K}$ is positive semi-definite if and only if $\mathbf{C} = \mathrm{Diag}(\mathbf{u})\,\mathbf{K}\,\mathrm{Diag}(\mathbf{u})$ is a positive semi-definite matrix for any $\mathbf{u} \neq 0$. Therefore the proof of the proposition 1 is equivalent to prove that $\mathbf{C} = \mathrm{Diag}(\mathbf{u})\left(\mathrm{Diag}\left(\frac{\mathbf{Bu}}{\mathbf{u}}\right) - \mathbf{B}\right)\mathrm{Diag}(\mathbf{u})$ is positive semi-definite for any $\mathbf{u} \neq 0$. Indeed, for any $\mathbf{h} \in \mathbb{R}^N$ with $\mathbf{h} \neq 0$ we have

$$
\begin{aligned}
\mathbf{h}^\top \mathbf{C} \mathbf{h} &= \sum_{1 \leq n \leq N} (\mathbf{Bu})_n u_n h_n^2 - \sum_{1 \leq n,m \leq N} b_{n,m} u_n u_m h_n h_m \\
&= \sum_{1 \leq n,m \leq N} b_{n,m} u_n u_m h_n^2 - \sum_{1 \leq n,m \leq N} b_{n,m} u_n u_m h_n h_m \\
&= \sum_{1 \leq n,m \leq N} b_{n,m} u_n u_m \frac{(h_n - h_m)^2}{2} \geq 0.
\end{aligned}
$$

This concludes the proof. ∎

## A.2 Proof of Proposition 2 (closed-form majorant)

We will prove this proposition by contradiction. Assume that $\mathbf{u}^*$ is a solution to the optimization problem, $\min_{\mathbf{u}>0} \varphi(\mathbf{u})$, with $\varphi(\mathbf{u}) = \left\|\frac{\mathbf{Bu}}{\mathbf{u}}\right\|_1$, and suppose that there exist indices $1 \le \bar{i}, \bar{j} \le R$ such that $u_{\bar{i}}^* \ne u_{\bar{j}}^*$. We will show that it is possible to construct another positive vector $\bar{\mathbf{u}} \in \mathbb{R}^R$ such that $\varphi(\bar{\mathbf{u}}) < \varphi(\mathbf{u}^*)$, thereby contradicting the optimality of $\mathbf{u}^*$.

Define $\bar{\mathbf{u}}$ by:

$$\bar{u}_r = \begin{cases} u_r^* & \text{if } r \notin \{\bar{i}, \bar{j}\}, \\ \dfrac{u_{\bar{i}}^* + u_{\bar{j}}^*}{2} & \text{if } r \in \{\bar{i}, \bar{j}\}. \end{cases}$$

Since $\mathbf{B}$ is a symmetric matrix with positive entries ($\mathbf{B} > 0$), and both $\mathbf{u}^*$ and $\bar{\mathbf{u}}$ are positive vectors, we can evaluate the objective function $\varphi$ as follows:

$$\begin{aligned}
\varphi(\bar{\mathbf{u}}) &= \sum_{\substack{1 \le i,j \le R \\ \{i,j\} \ne \{\bar{i},\bar{j}\}}} \frac{b_{j,i} u_i^*}{u_j^*} + b_{\bar{i},\bar{j}} \left( \frac{\bar{u}_{\bar{i}}}{\bar{u}_{\bar{j}}} + \frac{\bar{u}_{\bar{j}}}{\bar{u}_{\bar{i}}} \right) \\
&= \sum_{\substack{1 \le i,j \le R \\ \{i,j\} \ne \{\bar{i},\bar{j}\}}} \frac{b_{j,i} u_i^*}{u_j^*} + 2 b_{\bar{i},\bar{j}} \\
&< \sum_{\substack{1 \le i,j \le R \\ \{i,j\} \ne \{\bar{i},\bar{j}\}}} \frac{b_{j,i} u_i^*}{u_j^*} + b_{\bar{i},\bar{j}} \left( \frac{u_{\bar{i}}^*}{u_{\bar{j}}^*} + \frac{u_{\bar{j}}^*}{u_{\bar{i}}^*} \right) \varphi(\mathbf{u}^*),
\end{aligned}$$

where the strict inequality follows from the fact that for $a \ne b > 0$, we have $\frac{a}{b} + \frac{b}{a} > 2$. This contradicts the assumption that $\mathbf{u}^*$ is a minimizer of $\varphi$. Therefore, the optimal solution must satisfy $u_i^* = u_j^*$ for all $i, j$, i.e., $\mathbf{u}^*$ must be a constant vector. ∎

## B  Optimality of the Min-Max Problem Solution

This appendix shows that any eigenvector associated with the largest singular value of the symmetric matrix $\mathbf{B} > 0$ is a solution to the min-max problem:

$$\arg\min_{\mathbf{u}>0} \left\| \frac{\mathbf{B}^\top \mathbf{u}}{\mathbf{u}} \right\|_\infty \tag{29}$$

Assume that $\mathbf{u}^*$ is a solution of Problem equation 29. Then, we claim that for all $1 \le r \le R$,

$$\frac{\mathbf{b}_r^\top \mathbf{u}^*}{u_r^*} = t^* = \min_{\mathbf{u}>0} \left\| \frac{\mathbf{B}^\top \mathbf{u}}{\mathbf{u}} \right\|_\infty .$$

Suppose instead that this equality does not hold. Then, $(\mathbf{u}^*, t^*)$ cannot be optimal. Indeed, define the index set

$$I = \left\{ r \in \{1, \dots, R\} : \frac{\mathbf{b}_r^\top \mathbf{u}^*}{u_r^*} = t^* \right\},$$

and assume its complement $I^c = \{1, \dots, R\} \setminus I$ is non-empty. Then

$$\frac{\mathbf{b}_i^\top \mathbf{u}^*}{u_i^*} > \frac{\mathbf{b}_r^\top \mathbf{u}^*}{u_r^*}, \quad \forall i \in I^c, \forall r \in I.$$

We now construct a perturbed vector $\bar{\mathbf{u}} > 0$ such that $\left\| \frac{\mathbf{B}^\top \bar{\mathbf{u}}}{\bar{\mathbf{u}}} \right\|_\infty < t^*$, leading to a contradiction. Let $\bar{\mathbf{u}} = (\bar{u}_i)_{1 \le i \le R}$ be defined as a function of a positive scalar $\xi$ by

$$\bar{u}_i = \begin{cases} u_i^*, & \text{if } i \in I, \\ u_i^* - \xi, & \text{if } i \in I^c. \end{cases}$$

We now determine conditions under which $\left\| \frac{\mathbf{B}^\top \bar{\mathbf{u}}}{\bar{\mathbf{u}}} \right\|_\infty < t^*$. This is equivalent to finding $\xi > 0$ such that:

$$\text{If } \xi < u^*_{\min} := \min_{1 \leq r \leq R} u^*_r \tag{30a}$$

$$\text{For } r \in I : \quad \frac{\mathbf{b}_r^\top \mathbf{u}^*}{u^*_r} - \frac{\xi}{u^*_r} \sum_{\ell \in I^c} b_{r,\ell} < t^* \quad \Longleftrightarrow \quad \xi > 0 \tag{30b}$$

$$\text{For } r \in I^c : \quad \frac{\mathbf{b}_r^\top \mathbf{u}^*}{u^*_r - \xi} - \frac{\xi}{u^*_r - \xi} \sum_{\ell \in I^c} b_{r,\ell} < t^*$$

$$\Longleftrightarrow \quad \xi \left( t^* - \sum_{\ell \in I^c} b_{r,\ell} \right) < u^*_r t^* - \mathbf{b}_r^\top \mathbf{u}^* \tag{30c}$$

For any $r \in I^c$ we have $u^*_r t^* - \mathbf{b}_r^\top \mathbf{u}^* > 0$, now denotes $J = \{ r \in I^c : t^* - \sum_{\ell \in I^c} b_{r,\ell} \leq 0 \}$

$$\text{Eq. } 30c \Longleftrightarrow \begin{cases} \xi \geq 0 & \text{if } J \neq \emptyset, \\ \xi < \vartheta = \min_{r \in I^c \setminus J} \frac{u^*_r t^* - \mathbf{b}_r^\top \mathbf{u}^*}{t^* - \sum_{\ell \in I^c} b_{r,\ell}} > 0 & \text{if } I^c \setminus J \neq \emptyset \end{cases} \tag{31}$$

Therefore combining equation 31 with equation 30a and equation 30b, we can choose $\xi$ satisfying:

$$\begin{cases} 0 < \xi < u^*_{\min} & \text{if } J \neq \emptyset, \\ 0 < \xi < \min(u^*_{\min}, \vartheta) & \text{if } I^c \setminus J \neq \emptyset \end{cases}$$

which contradicts the optimality of $(\mathbf{u}^*, t^*)$. Therefore, we must have: $\mathbf{B}\mathbf{u}^* = t^* \mathbf{u}^*$, which implies the eigenvalue problem:

$$(\mathbf{B} - t^* \mathbf{I})\mathbf{u}^* = \mathbf{0}.$$

We seek the largest eigenvalue $t^*$ of $\mathbf{B}$ such that the corresponding eigenvector $\mathbf{u}^*$ is nonnegative. By the Perron-Frobenius theorem, the eigenvector $\mathbf{z}$ associated with the largest eigenvalue of a positive matrix $\mathbf{B}$ is positive. Since all other eigenvectors are orthogonal to $\mathbf{z}$, they cannot satisfy the positivity constraint. Therefore, the only admissible solutions are of the form $\mathbf{u}^* = \alpha \mathbf{z}$ for $\alpha > 0$.
This confirms that:

$$\frac{\mathbf{B}\mathbf{u}^*}{\mathbf{u}^*} = \mu \mathbb{1}_R,$$

where $\mu$ is the largest eigenvalue of $\mathbf{B}$. AmSOM is then a simple scaled gradient descent algorithm.

## C   Proof of Theorems 3 and 5 (linear convergence)

### C.1   Proof of Linear Convergence Rate for Frobenius, with Noise

The proof relies on the fact that the update is a contraction of the form $\mathbf{I} - \gamma \operatorname{Diag}(\mathbf{B}\mathbb{1})^{-1}\mathbf{B}$. Matrices of form $\operatorname{Diag}(\mathbf{B}\mathbb{1})^{-1}\mathbf{B}$, which are commonly encountered in spectral graph theory (Brouwer & Haemers, 2012; Chung, 1997), are equivalent to the Laplacian matrices when $\operatorname{Diag}(\mathbf{B}\mathbb{1}) \geq 0$. These matrices have singular values bounded between 0 and 1. This can be shown from Proposition 1, setting $\mathbf{u}$ as the vector of ones.

**Lemma 1** *Let $\mathbf{B} \in \mathbb{R}^{r \times r}_+$ be an entry-wise positive and symmetric matrix. Then*

$$0 \preceq \operatorname{Diag}(\mathbf{B}\mathbb{1})^{-1}\mathbf{B} \preceq I.$$

We first exhibit the relationship between estimate $\mathbf{h}^{(k)}$ at iteration $k$ and the next iterate $\mathbf{h}^{(k+1)}$,

$$
\begin{aligned}
\left\| \mathbf{h}^{(k+1)} - \mathbf{h}^* \right\|_2 &= \left\| \left[ \mathbf{h}^{(k)} - \gamma \mathbf{A}(\mathbf{h}^{(k)})^{-1} \nabla \Psi(\mathbf{h}^{(k)}) \right]_\epsilon - \mathbf{h}^* \right\|_2 \\
&= \left\| \left[ \mathbf{h}^{(k)} - \gamma \left( \mathrm{Diag}(\mathbf{W}\mathbf{W}^\top \mathbb{1}) \right)^{-1} \left( \mathbf{W}\mathbf{W}^\top \mathbf{h}^{(k)} - \mathbf{W}\mathbf{v} \right) \right]_\epsilon - \mathbf{h}^* \right\|_2 \\
&= \left\| \left[ \mathbf{h}^{(k)} - \gamma \left( \mathrm{Diag}(\mathbf{W}\mathbf{W}^\top \mathbb{1}) \right)^{-1} \left( \mathbf{W}\mathbf{W}^\top \mathbf{h}^{(k)} - \mathbf{W}\mathbf{W}^\top \mathbf{h}^* - \mathbf{W}\boldsymbol{\eta} \right) \right]_\epsilon - \mathbf{h}^* \right\|_2 \\
&= \left\| \left[ \mathbf{h}^{(k)} - \gamma \left( \mathrm{Diag}(\mathbf{W}\mathbf{W}^\top \mathbb{1}) \right)^{-1} \left( \mathbf{W}\mathbf{W}^\top (\mathbf{h}^{(k)} - \mathbf{h}^*) - \mathbf{W}\boldsymbol{\eta} \right) \right]_\epsilon - \mathbf{h}^* \right\|_2 .
\end{aligned}
$$

where the diagonal elements of $\mathbf{A}(\mathbf{h}^{(k)})$ are nonzero because $\epsilon$ is positive, and we have used the decomposition of vector $\mathbf{v}$ as

$$
\mathbf{v} = \mathbf{W}^\top \mathbf{h}^* + \eta.
$$

The Karush-Kuhn-Tucker (KKT) conditions for the nonnegative least squares problem provide some useful additional constraints on the residuals vector $\eta$:

$$
\begin{cases}
\mathbf{W}[i,:]\eta = 0 & \text{if } \mathbf{h}^*[i] > \epsilon, \\
\mathbf{W}[i,:]\eta \le 0 & \text{if } \mathbf{h}^*[i] = \epsilon.
\end{cases}
$$

For any index $i$,

- either $\mathbf{h}[i]^* = \epsilon$ and the right-hand side summand at index $i$ is of the form $\left[ a + \frac{1}{b}\mathbf{W}[i,:]\eta \right]_\epsilon$ with positive $b$ and negative $\mathbf{W}[i,:]\eta$, which is upper bounded by $[a]_\epsilon$ for any real $a$.

- else $\mathbf{h}[i]^* > 0$ and $\mathbf{W}[i,:]\eta$ vanishes.

Therefore, we can upper-bound elementwise the estimation error

$$
\left\| \mathbf{h}^{(k+1)} - \mathbf{h}^* \right\|_2 \le \left\| \left[ \mathbf{h}^{(k)} - \gamma \left( \mathrm{Diag}(\mathbf{W}\mathbf{W}^\top \mathbb{1}) \right)^{-1} \mathbf{W}\mathbf{W}^\top \left( \mathbf{h}^{(k)} - \mathbf{h}^* \right) \right]_\epsilon - \mathbf{h}^* \right\|_2 .
$$

Denoting $\Pi_{\mathbf{h}^* - \epsilon}(\mathbf{h}) = \max(\mathbf{h}, -\mathbf{h}^* + \epsilon)$ the projection on the set of vectors with entries larger than $-\mathbf{h}^* + \epsilon$, we get

$$
\left\| \mathbf{h}^{(k+1)} - \mathbf{h}^* \right\|_2 \le \left\| \Pi_{\mathbf{h}^* - \epsilon} \left[ \left( \mathbf{I} - \gamma \left( \mathrm{Diag}(\mathbf{W}\mathbf{W}^\top \mathbb{1}) \right)^{-1} \mathbf{W}\mathbf{W}^\top \right) \left( \mathbf{h}^{(k)} - \mathbf{h}^* \right) \right] \right\|_2 .
$$

The solution $\mathbf{h}^*$ must be feasible; therefore, the difference $-\mathbf{h}^* + \epsilon$ is always elementwise negative. Consequently, the projector $\Pi_{\mathbf{h}^* - \epsilon}$ is a contraction. Combined with the spectral norm inequality,

$$
\left\| \mathbf{h}^{(k+1)} - \mathbf{h}^* \right\|_2 \le \left\| \mathbf{I} - \gamma \left( \mathrm{Diag}(\mathbf{W}\mathbf{W}^\top \mathbb{1}) \right)^{-1} \mathbf{W}\mathbf{W}^\top \right\|_2 \left\| \mathbf{h}^{(k)} - \mathbf{h}^* \right\|_2 ,
$$

where $\|\mathbf{M}\|_2$ is the spectral norm of matrix $\mathbf{M}$.

Denoting $\mathbf{B} := \mathbf{W}\mathbf{W}^\top$, according to Lemma 1, the singular values of $\gamma(\mathrm{Diag}(\mathbf{B}\mathbb{1}))^{-1}\mathbf{B}$ live in the open interval $(0, \gamma)$. Consequently, matrix $\mathbf{I} - \gamma(\mathrm{Diag}(\mathbf{W}\mathbf{W}^\top \mathbb{1}))^{-1}\mathbf{W}\mathbf{W}^T$ has singular values in $(1 - \gamma, 1)$. Linear convergence for the iterates of mSOM is obtained as follows:

$$
\frac{\|\mathbf{h}^{(k+1)} - \mathbf{h}^*\|_2}{\|\mathbf{h}^{(k)} - \mathbf{h}^*\|_2} \le \left\| (\mathbf{I} - \gamma \, \mathrm{Diag}((\mathbf{W}\mathbf{W}^\top \mathbb{1})^{-1})\mathbf{W}\mathbf{W}^\top) \right\|_2 = \mu
$$

with $0 < \mu < 1$ when $|1 - \gamma| < 1$, *i.e.* when $0 < \gamma < 2$.

## C.2 Convergence Proof for Local Linear Rate with $\beta$-divergence and KL

We focus only on the convergence rate of the proposed algorithm for sub-problems without error. The proof idea is similar to the Frobenius case, albeit more technical. We start by rewritting the update rules as follows when $\Psi$ is the $\beta$ divergence with $\beta \in [1, 2]$,

$$
\begin{aligned}
\left\| \mathbf{h}^{(k+1)} - \mathbf{h}^* \right\|_2 &= \left\| \left[ \mathbf{h}^{(k)} - \gamma \mathbf{A}(\mathbf{h}^{(k)})^{-1} \Big( \sum_{m=1}^{M} (\mathbf{w}_m^\top \mathbf{h}^{(k)})^{\beta-1} \mathbf{w}_m - v_m (\mathbf{w}_m^\top \mathbf{h}^k)^{\beta-2} \mathbf{w}_m \Big) \right]_\epsilon - \mathbf{h}^* \right\|_2 \\
&= \left\| \left[ \mathbf{h}^{(k)} - \gamma \mathbf{A}(\mathbf{h}^{(k)})^{-1} \Big( \sum_{m=1}^{M} (\mathbf{w}_m^\top \mathbf{h}^{(k)})^{\beta-1} \mathbf{w}_m - \mathbf{w}_m^\top \mathbf{h}^* (\mathbf{w}_m^\top \mathbf{h}^{(k)})^{\beta-2} \mathbf{w}_m \Big) \right]_\epsilon - \mathbf{h}^* \right\|_2 \\
&= \left\| \Pi_{\mathbf{h}^*-\epsilon} \left[ \Big( \mathbf{I} - \gamma \mathbf{A}(\mathbf{h}^{(k)})^{-1} \sum_{m=1}^{M} (\mathbf{w}_m^\top \mathbf{h}^{(k)})^{\beta-2} \mathbf{w}_m \mathbf{w}_m^\top \Big) (\mathbf{h}^{(k)} - \mathbf{h}^*) \right] \right\|_2
\end{aligned}
$$

with

$$
\begin{aligned}
\mathbf{A}(\mathbf{h}^{(k)}) &= \mathrm{Diag} \Big( \sum_{m=1}^{M} \Big( (\beta-1)(\mathbf{w}_m^\top \mathbf{h}^{(k)})^{\beta-2} - (\beta-2) v_m (\mathbf{w}_m^\top \mathbf{h}^{(k)})^{\beta-3} \Big) \mathbf{w}_m \mathbf{w}_m^\top \mathbb{1} \Big) \\
&= \mathrm{Diag} \Big( \sum_{m=1}^{M} \mathbf{w}_m^\top \Big( (\beta-1) \mathbf{h}^{(k)} - (\beta-2) \mathbf{h}^* \Big) (\mathbf{w}_m^\top \mathbf{h}^{(k)})^{\beta-3} \mathbf{w}_m \mathbf{w}_m^\top \mathbb{1} \Big),
\end{aligned}
$$

where we have assumed that $v_m = \mathbf{w}_m^T \mathbf{h}^*$. For nonnegative $\mathbf{h}^* - \epsilon$ the projection $\Pi_{\mathbf{h}^*-\epsilon}$ is a contraction, therefore

$$
\left\| \mathbf{h}^{(k+1)} - \mathbf{h}^* \right\|_2 \le \left\| \mathbf{I} - \gamma \mathbf{A}(\mathbf{h}^{(k)})^{-1} \sum_{m=1}^{M} (\mathbf{w}_m^\top \mathbf{h}^{(k)})^{\beta-2} \mathbf{w}_m \mathbf{w}_m^\top \right\|_2 \left\| \mathbf{h}^{(k)} - \mathbf{h}^* \right\|_2 .
$$

Since $\mathbf{A}(\mathbf{h}^{(k)}) \succ 0$, we now aim to bound the singular values of the matrix multiplied by the $\gamma$ factor. Suppose there exist two matrices $\mathbf{Z}_1$ and $\mathbf{Z}_2$ such that

$$
0 \preceq \mathbf{Z}_1 \prec \mathbf{A}(\mathbf{h}^{(k)})^{-1} \sum_{m=1}^{M} (\mathbf{w}_m^\top \mathbf{h}^{(k)})^{\beta-2} \mathbf{w}_m \mathbf{w}_m^\top \prec \mathbf{Z}_2 \preceq \frac{2}{\gamma} \mathbf{I}. \tag{32}
$$

Under such conditions, we get that

$$
-\mathbf{I} \preceq \mathbf{I} - \gamma \mathbf{Z}_2 \prec \mathbf{I} - \gamma \mathbf{A}(\mathbf{h}^{(k)})^{-1} \sum_{m=1}^{M} (\mathbf{w}_m^\top \mathbf{h}^{(k)})^{\beta-2} \mathbf{w}_m \mathbf{w}_m^\top \prec \mathbf{I} - \gamma \mathbf{Z}_1 \preceq \mathbf{I},
$$

Then we will conclude that

$$
\left\| \mathbf{I} - \gamma \mathbf{A}(\mathbf{h}^{(k)})^{-1} \sum_{m=1}^{M} (\mathbf{w}_m^\top \mathbf{h}^{(k)})^{\beta-2} \mathbf{w}_m \mathbf{w}_m^\top \right\|_2 \le \max_{i \in \{1,2\}} \left\| \mathbf{I} - \gamma \mathbf{Z}_i \right\|_2 .
$$

We may then bound the error decrease at each iteration if $\mathbf{Z}_1, \mathbf{Z}_2$ follow the general form of Lemma 1. To build these matrices $\mathbf{Z}_1, \mathbf{Z}_2$, but also to upper-bound the singular values by one, we rely on the following Lemma.

**Lemma 2** *Let $\mathbf{D}_1, \mathbf{D}_2 \in \mathbb{R}^{n \times n}$ two diagonal matrices, such that $\mathbf{D}_1 \prec \mathbf{D}_2$. Then for any positive definite or positive semi-definite matrix $\mathbf{M} \in \mathbb{R}^{n \times n}$, $\mathbf{D}_1 \mathbf{M} \prec \mathbf{D}_2 \mathbf{M}$ or $\mathbf{D}_1 \mathbf{M} \preceq \mathbf{D}_2 \mathbf{M}$, respectively.*

Therefore, our goal will be to build diagonal matrices $\mathbf{Z}_1$ and $\mathbf{Z}_2$. For $i \in \{1, 2\}$, let

$$\mathbf{A}_i = \alpha_i \operatorname{Diag}(\sum_{m=1}^{M} (\mathbf{w}_m^\top \mathbf{h}^{(k)})^{\beta-2} \mathbf{w}_m \mathbf{w}_m^\top \mathbb{1}), \tag{33}$$

such that, $0 \prec \mathbf{A}_2 \prec \mathbf{A}(\mathbf{h}^{(k)}) \prec \mathbf{A}_1$. Then we can set

$$\mathbf{Z}_i = \mathbf{A}_i^{-1} \sum_{m=1}^{M} (\mathbf{w}_m^\top \mathbf{h}^{(k)})^{\beta-2} \mathbf{w}_m \mathbf{w}_m^\top$$

and Lemma 2 ensures that $\mathbf{Z}_1 \preceq \mathbf{A}(\mathbf{h}^{(k)})^{-1} \sum_{m=1}^{M} (\mathbf{w}_m^\top \mathbf{h}^{(k)})^{\beta-2} \mathbf{w}_m \mathbf{w}_m^\top \preceq \mathbf{Z}_2$. Since Lemma 1 imposes that $\mathbf{Z}_i$ has its spectrum in $[0, \frac{1}{\alpha_i}]$, to ensure $\max_{i \in \{1,2\}} \|\mathbf{I} - \gamma \mathbf{Z}_i\|_2 \leq 1$ we need to find $\alpha_1$ and $\alpha_2$ such that $0 \prec \mathbf{A}_2 \prec \mathbf{A}(\mathbf{h}^{(k)}) \prec \mathbf{A}_1$ and

$$\alpha_1 > \alpha_2 \geq \gamma/2.$$

To satisfy the first condition, it is sufficient that for $1 \leq m \leq M$, we have

$$\alpha_2 \mathbf{w}_m^\top \mathbf{h}^{(k)} \leq (\beta - 1) \mathbf{w}_m^\top \mathbf{h}^{(k)} - (\beta - 2) \mathbf{w}_m^\top \mathbf{h}^* \leq \alpha_1 \mathbf{w}_m^\top \mathbf{h}^{(k)}$$

$$\iff 1 - \frac{1 - \alpha_2}{2 - \beta} \leq \frac{\mathbf{w}_m^\top \mathbf{h}^*}{\mathbf{w}_m^\top \mathbf{h}^{(k)}} \leq 1 + \frac{\alpha_1 - 1}{2 - \beta}.$$

This last constraint can be achieved by assuming that there exists $0 \leq \nu < 1$ such that $1 - \nu \leq \frac{\mathbf{w}_m^\top \mathbf{h}^*}{\mathbf{w}_m^\top \mathbf{h}^{(k)}} \leq 1 + \nu$ for all $m \leq M$ and we denote this condition by $(\mathcal{H}_k)$. Indeed, we may set $\alpha_1$ and $\alpha_2$ such that, $\gamma/2 \leq \alpha_2 \leq 1 - \nu(2 - \beta) \leq 1 + \nu(2 - \beta) \leq \alpha_1$. To conclude the proof of convergence, we need to show that condition $(\mathcal{H}_k)$ holds for all iterations under the assumptions of Theorem 5.

We finish the proof by showing that if for all $m \leq M$ there exists $0 \leq \eta \leq 1$, such that the convergence condition is satisfied:

$$|\mathbf{w}_m^\top \mathbf{h}^{(0)} - \mathbf{w}_m^\top \mathbf{h}^*| \leq \eta \mathbf{w}_m^\top \mathbf{h}^* - 2\|\mathbf{w}_m\|_2 \|\mathbf{h}^{(0)} - \mathbf{h}^*\|_2, \tag{$\mathcal{P}_0$}$$

and if the stepsize $\gamma$ is chosen to ensure the decrease of the estimation error as shown above, then $(\mathcal{H}_k)$ is true for all $k \geq 0$.

First note that $(\mathcal{P}_0)$ implies that $(\mathcal{H}_0)$. Indeed $(\mathcal{P}_0)$ implies $|\mathbf{w}_m^\top \mathbf{h}^{(0)} - \mathbf{w}_m^\top \mathbf{h}^*| \leq \eta \mathbf{w}_m^\top \mathbf{h}^{(0)}$ which is equivalent to $1 - \frac{\eta}{1+\eta} \leq \frac{\mathbf{w}_m^\top \mathbf{h}^*}{\mathbf{w}_m^\top \mathbf{h}^{(0)}} \leq 1 + \frac{\eta}{1-\eta}$ for all $m \leq M$ and nonnegative $\mathbf{w}_m$ and $\mathbf{h}^{(0)}$. We can choose $\nu = \max(\frac{\eta}{1-\eta}, \frac{\eta}{1+\eta}) = \frac{\eta}{1-\eta}$ so that $(\mathcal{H}_0)$ is verified. We are going to show that with this choice of $\nu$, $(\mathcal{H}_j)$ is satisfied for all $k \geq 0$.

Now suppose $(\mathcal{P}_0)$ and $(\mathcal{H}_k)$ for all $j \leq k$. Let us prove that $(\mathcal{H}_{k+1})$ holds. For any $m \leq M$, by triangular inequality,

$$\left| \mathbf{w}_m^\top \mathbf{h}^* - \mathbf{w}_m^\top \mathbf{h}^{(k+1)} \right| \leq \left| \mathbf{w}_m^\top \mathbf{h}^* - \mathbf{w}_m^\top \mathbf{h}^{(0)} \right| + \left| \mathbf{w}_m^\top \left( \mathbf{h}^{(0)} - \mathbf{h}^{(k+1)} \right) \right|.$$

Using $(\mathcal{P}_0)$, the first term is bounded from above:

$$|\mathbf{w}_m^\top \mathbf{h}^* - \mathbf{w}_m^\top \mathbf{h}^{(0)}| \leq \eta \mathbf{w}_m^\top \mathbf{h}^* - 2\|\mathbf{w}_m\|_2 \|\mathbf{h}^{(0)} - \mathbf{h}^*\|_2.$$

The second term can be bounded in absolute value using the Cauchy-Schwarz and triangular inequalities:

$$\left| \mathbf{w}_m^\top \left( \mathbf{h}^{(k+1)} - \mathbf{h}^{(0)} \right) \right| \leq \|\mathbf{w}_m\|_2 \|\mathbf{h}^{(k+1)} - \mathbf{h}^{(0)}\|_2$$

$$\leq \|\mathbf{w}_m\|_2 \left( \|\mathbf{h}^{(k+1)} - \mathbf{h}^*\|_2 + \|\mathbf{h}^{(0)} - \mathbf{h}^*\|_2 \right).$$

Because $(\mathcal{H}_j)$ is verified for any $j \leq k$, if $\gamma$ satisfies the condition equation C.2, we know that the mSOM iteration diminishes the distance to the solution for all iterations up to $k + 1$. Therefore,

$$\|\mathbf{h}^{(k+1)} - \mathbf{h}^*\|_2 \leq \|\mathbf{h}^{(k)} - \mathbf{h}^*\|_2 \leq \|\mathbf{h}^{(0)} - \mathbf{h}^*\|_2.$$

Putting it all together, we obtain the desired inequality,

$$\left| \mathbf{w}_m^\top (\mathbf{h}^* - \mathbf{h}^{(k+1)}) \right| \leq \eta \mathbf{w}_m^\top \mathbf{h}^* - 2\|\mathbf{w}_m\|_2 \|\mathbf{h}^{(0)} - \mathbf{h}^*\|_2 + 2\|\mathbf{w}_m\|_2 \|\mathbf{h}^{(0)} - \mathbf{h}^*\|_2 \leq \eta \mathbf{w}_m^\top \mathbf{h}^*.$$

Again, for $\nu = \max(\frac{\eta}{1-\eta}, \frac{\eta}{1+\eta})$, $(\mathcal{H}_{k+1})$ is verified. ∎

# D  Proof of Proposition 6 (optimal scaling)

The loss $D_\beta$ is separable. We can therefore compute the optimal scaling $\lambda$ for a single column $\mathbf{h}$ of $\mathbf{H}$ corresponding to column $\mathbf{v}$ of the data matrix $\mathbf{V}$. Noting that

$$D_\beta(\mathbf{v}, \lambda\mathbf{Wh}) = \sum_{m=1}^{M} \frac{1}{\beta(\beta-1)}\big((\beta-1)\lambda^\beta(\mathbf{w}_m^\top\mathbf{h})^\beta - \beta\lambda^{\beta-1}v_m(\mathbf{w}_m^\top\mathbf{h})^{\beta-1}\big) + C$$

where $C$ is a constant independent of $\lambda$. Setting the derivative of $D_\beta$ to zero yields a single solution

$$\lambda^* = \frac{\sum_{m=1}^{M} v_m(\mathbf{w}_m^\top\mathbf{h})^{\beta-1}}{\sum_{m=1}^{M}(\mathbf{w}_m^\top\mathbf{h})^\beta},$$

which is always positive. We assumed implicitly that at least one $\mathbf{w}_m^\top\mathbf{h}$ is positive, which happens as long as both $\mathbf{W}$ and $\mathbf{H}$ are positive. ∎

# E  Additional Figures

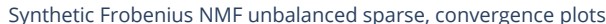

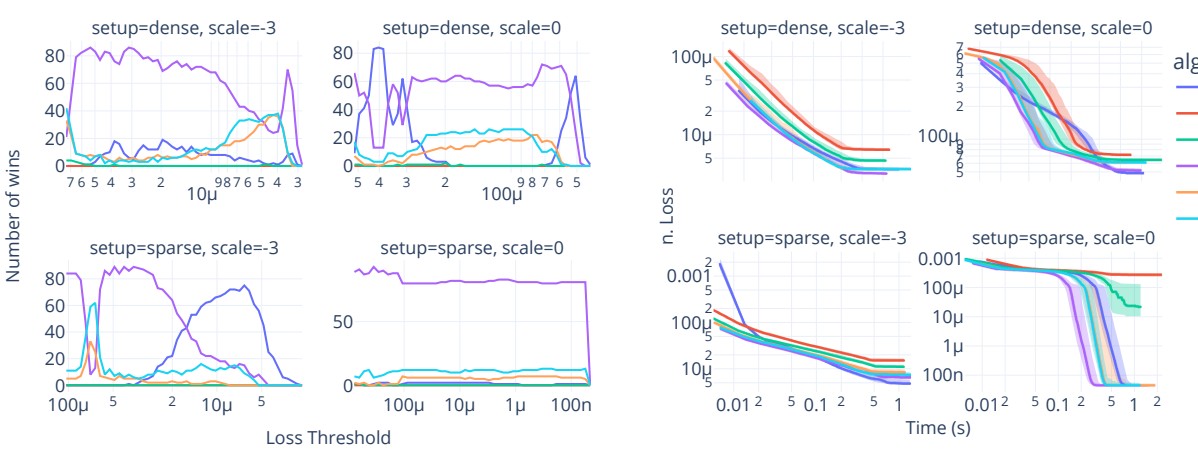

Figure 6: Median plots for the synthetic experiments on the unbalanced and sparse NMF problem with the Frobenius loss. The dimensions are $[M, N, R] = [1000, 400, 20]$. The maximal number of iterations is $K_{\max} = 200$. Error bands show the 0.25 and 0.75 quartile errors. In the scaled experiment, the columns of $\mathbf{W}$, which have been scaled on a logarithmic scale from 1 to $10^{-3}$. The sparse setup is similar to what is described for the KL-divergence experiment in section 5.

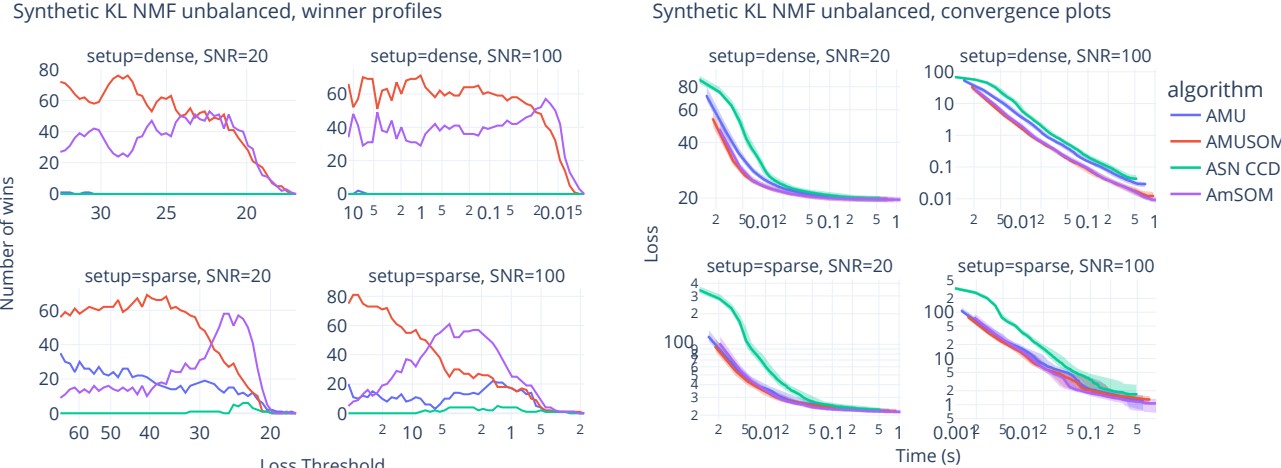

Figure 7: Median plots for the synthetic experiments on the unbalanced NMF problem with the KL-divergence loss. The dimensions are $[M, N, R] = [200, 100, 10]$. The maximal number of iterations is $K_{\max} = 400$ (80 for ASN CCD). Error bands show the 0.25 and 0.75 quartile errors. The Poisson noise intensity has been adjusted to account for the balancing of the columns of $\mathbf{W}$, which have been scaled on a logarithmic scale from 1 to $10^{-3}$.

