# Supplemental material for
# A Second Order Majorant Algorithm
# for Nonnegative Matrix Factorization

In this document, we provide additional figures and geometric examples, as well as supplementary experiments, to address the reviewers' questions. Since several questions raised by different reviewers overlap, we group them by topic. For each topic, we explicitly recall the corresponding reviewer questions before presenting our responses.

## 1   Why using $\ell_1$-norm to optimize the positive vector $\mathbf{u}$ ? (R1, Q4 and R2, Q1)

The goal of the proposed second-order type algorithm is to accelerate convergence while maintaining low computational complexity. In this sense, the proposed method indeed follows a MM strategy, although with a specific focus on constructing efficient diagonal second-order surrogates.

More precisely, Proposition 1 shows that a family of diagonal matrices of the form $\mathrm{Diag}\left(\frac{\mathbf{Bu}}{\mathbf{u}}\right)$, for $\mathbf{u} > 0$, where $\mathbf{B}$ denotes the true Hessian matrix of the loss, satisfies a majorization constraint. This result also explains why multiplicative updates naturally arise in this framework, as they correspond to particular choices of the vector $\mathbf{u}$.

Motivated by this observation, our objective is to select a positive vector $\mathbf{u}$ such that the resulting diagonal majorant is as close as possible to the true Hessian, while remaining computationally efficient. A natural first idea would be to minimize the $\ell_2^2$ distance between them, which can be formulated as follows:

$$\mathbf{u} \in \arg\min_{\mathbf{u} \geq \epsilon} \left\|\mathrm{Diag}\left(\frac{\mathbf{Bu}}{\mathbf{u}}\right) - \mathbf{B}\right\|_2^2 \tag{1}$$

This sub-optimization problem does not have a closed-form solution. Moreover, we can see that it forces the diagonal majorant to the diagonal of the true Hessian. We then decided to use $\ell_1$-norm instead of $\ell_2^2$, i.e.

$$\mathbf{u} \in \arg\min_{\mathbf{u} \geq \epsilon} \left\|\mathrm{Diag}\left(\frac{\mathbf{Bu}}{\mathbf{u}}\right) - \mathbf{B}\right\|_1 = \arg\min_{\mathbf{u} \geq \epsilon} \sum_{r=1}^{R} \left|\frac{\mathbf{b}_r^\top \mathbf{u}}{u_r} - b_{r,r}\right| = \arg\min_{\mathbf{u} \geq \epsilon} \sum_{r=1}^{R} \frac{\mathbf{b}_r^\top \mathbf{u}}{u_i} \tag{2}$$

which equivalent to the Eq. (9) in the manuscript. This last equality is due to the fact that both $\mathbf{B}$ and $\mathbf{u}$ are positive, and that

$$\frac{\mathbf{b}_r^\top \mathbf{u}}{u_r} - b_{r,r} = \sum_{j \neq r} \frac{b_{rj} u_j}{u_r} \geq 0.$$

To illustrate the behavior of the two strategies, we consider the following quadratic problem:

$$\mathbf{x} = \arg\min_{\mathbf{x} \geq \epsilon} \left(f(\mathbf{x}) = \frac{1}{2}\mathbf{x}^\top \mathbf{H}\mathbf{x} + \mathbf{b}^\top \mathbf{x}\right) \tag{3}$$

where $\mathbf{H}$ is a symmetric matrix with strictly positive entries. We set the dimension of variable to 200 (i.e., $\mathbf{x} \in \mathbb{R}_{++}^{200}$). We denote $\mathbf{x}^*$ as the optimizer solution of the problem. We generate three instances of $\mathbf{H}$ corresponding to different distributions of its diagonal values:

1. **Small variance case (Fig. 1)** The diagonal entries of $\mathbf{H}$ are concentrated in a narrow range. In this regime, both $\ell_1$-based and $\ell_2$-based majorants behave similarly, although the $\ell_1$-based method remains slightly faster due to the absence of an inner optimization loop.

2. **Moderate variance case (Fig. 2)** The diagonal entries exhibit moderate dispersion. In this setting, the $\ell_1$-based majorant clearly outperforms the $\ell_2$-based one. The $\ell_2$ formulation tends to overfit large diagonal values, leading to conservative steps and slower convergence.

3. **High variance (ill-conditioned) case (Fig. 3)** The diagonal entries span several orders of magnitude. This situation highlights the main advantage of the $\ell_1$ formulation: it is significantly more robust to scale heterogeneity and avoids domination by large curvature directions.

Figures 1, 2, and 3 illustrate the three considered cases of the matrix $\mathbf{H}$ (left), as described above, together with the corresponding evolution of the objective function (right). The horizontal axis represents the iteration index, while the vertical axis shows the objective gap $f(\mathbf{x}_k) - f(\mathbf{x}^*)$. The blue curve corresponds to the iterates $x_k$ obtained using the $\ell_1$-based majorant, whereas the red curve corresponds to the $\ell_2$-based majorant.

A first observation is that the method based on the $\ell_2$-norm is significantly more time-consuming, due to the need to solve an additional sub-optimization problem to compute the majorant, whereas the $\ell_1$-based formulation admits a closed-form solution.

A second observation is that a noticeable difference between the two approaches appears mainly when the matrix $\mathbf{H}$ is highly ill-conditioned (Case 3). As shown in Fig. 3, the $\ell_1$-based method converges much faster and remains numerically stable, while the $\ell_2$-based approach suffers from slower convergence and higher computational cost.

For clarity of visualization, all curves are displayed using a logarithmic scale.

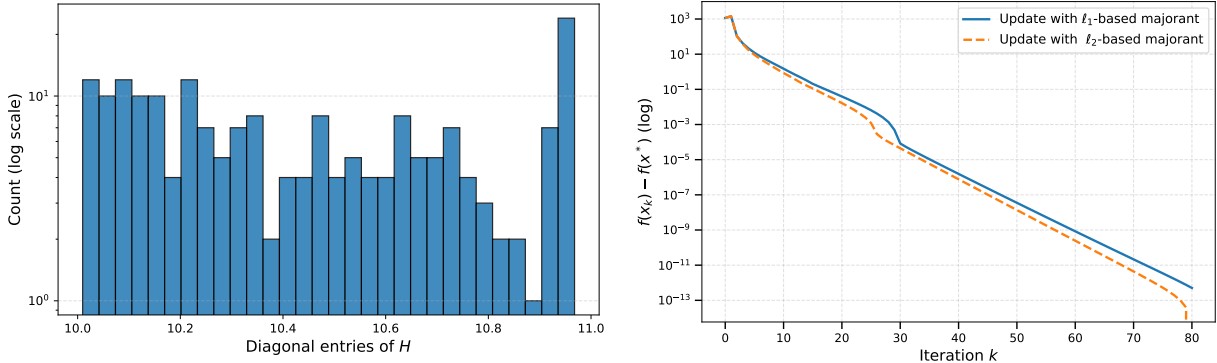

Figure 1: **Runtime with: $\ell_1$-based: 0.001 s. and $\ell_2$-based: 0.401 s.**

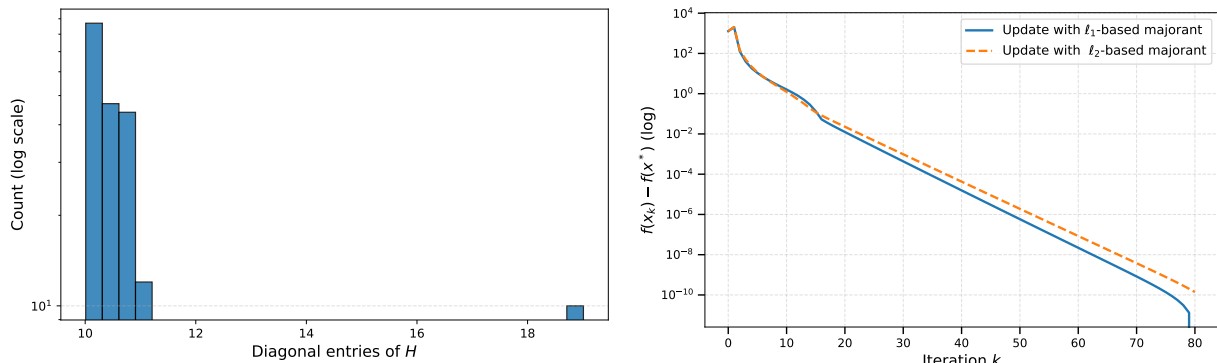

Figure 2: **Runtime with: $\ell_1$-based: 0.002 s. and $\ell_2$-based: 0.569 s.**

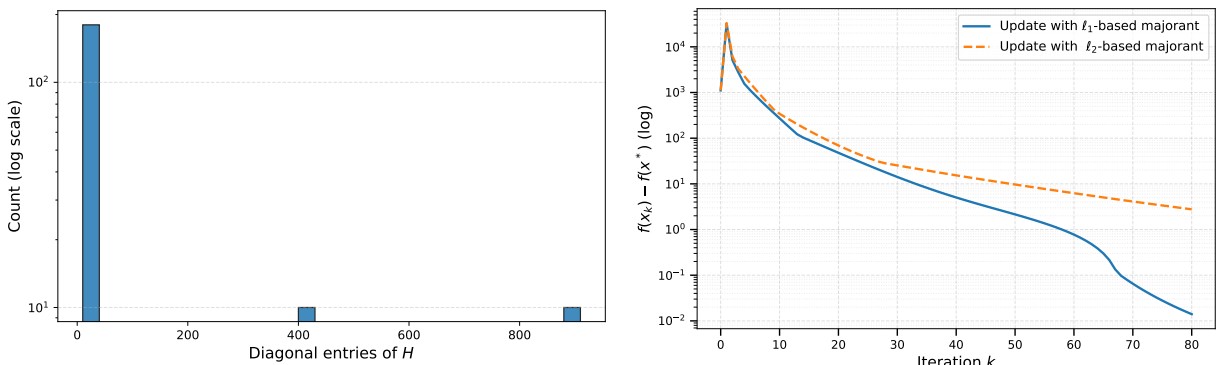

Figure 3: **Runtime with: $\ell_1$-based: 0.001 s. and $\ell_2$-based: 0.335 s.**