# OpenReview forum: "A Second Order Majorant Algorithm for Nonnegative Matrix Factorization"
_TMLR — Accepted by TMLR_

### Review · Reviewer_3f7V · 2025-11-24

**Summary Of Contributions:**

This paper proposes a novel framework called second order majorization to compute low-rank matrix factorization with nonnegativity constraints. Alternating optimization (AO) is an effective optimization method to obtain a low-rank factorization (with or without nonnegativity constraints) of the desired matrix. AO solves the problem iteratively by optimizing the loss with respect to one parameter matrix while keeping the other fixed. This results in convex subproblems (with nonnegativity constraints) that can be solved via multiple methods. The proposed framework proposes a Newton's-like method where the Hessian is approximated to be diagonal with positive entries. The authors show how to derive a family of diagonal Hessian approximations which locally upper bound the original cost function and also include the well-known MWU method. The authors propose an approximation that is computationally efficient and theoretically rigorous to guarantee local linear convergence for Frobenius and $\beta$-divergence loss. Authors also propose an improvement to the MU method and analyze the performance of various algorithms for synthetic as well as real world matrices.

Computing an NMF is a well-studied and important problem and this paper provides a new perspective on deriving new update rules as well as improving the existing ones. The framework proposed in the paper is theoretically sound with neat proofs and provides new insights for the existing MU method. However, it is not clear from the numerical experiments if this method is significantly better than the state-of-the-art for real-world matrices. There is also no insight provided for what kind of matrices the new update might be better than the existing MU update (there might be some, see suggestions).

In my opinion, the motivation and algorithms also misses a key point that needs to be highlighted in the proposed framework (see suggestions below).

**Additional Comments:**

Small corrections below

Proposition $1$: $\mathbf B \in \mathbb{R}_{+}$, no need to say nonnegative entries.

Proof of proposition $1$: replace $x$ by $h$.

Brackets appear as $] \dots [$ instead of $[  \dots ]$

**Audience:**

Yes

**Audience Explanation:**

The paper provides new updates for alternating optimization algorithm for NMF. NMF is a critical part of ML pipelines for analyzing and performing downstream tasks on nonnegative matrices. This manuscript also provides some insights to construct new updates for the same problem, and may be useful for nonnegative tensor factorization (used in multi-dimensional data analysis) as well.

**Broader Impact Concerns:**

No concerns

**Claims And Evidence:**

Yes

**Claims Explanation:**

Authors claim that their method converges linearly which is theoretically proved.

I am slightly concerned about the statement that (A)mSOM outperforms the state-of-the-art often. I think mSOM is competitive with SOTA, and MUSOM (which is partly previous work, partly proposed as the new framework provides motivation for scaling the MU updates) outperforms in some of the cases.

**Requested Changes:**

The paper is well-structured and well-written with neat proofs. I enjoyed the reading. I do have some suggestions that would make the paper stronger in my opinion.

1.) This is one of the most important points, please make the plots larger, with legends clearer (use more differentiable linestyle/markers). Also, it would be good to see the variance of different algorithms. The current plots only include medians.

2.) The mSOM algorithm appears to be doing well for small noise (almost exact matrix-factorization), but is not too different from others when noise is added (which is more practical).
I propose a synthetic experiment that might provide some insight in what cases mSOM might be better than MU:
Since the majorization focuses on minimizing the $1$-norm of the diagonal matrices, it would be good to see what happens if the constructed synthetic matrices have differences in scales (happens in practical scenarios), i.e., scale the columns of the factors by large amounts making the underlying Hessian somewhat ill-conditioned. It is likely that mSOM does better than MU in this case with some intuition why that might be the case.

3.) This is in relation to the above experiment. Note that NMF is non-unique, i.e., I can absorb diagonal scaling and its inverse to both the factor rows without changing the factorization. When alternating optimization is performed, it choses `an' element of the equivalence class of all the non-unique factorizations and optimizes things. This may work ok in practice, but might or might not get into trouble for ill-conditioned Hessians. One nice thing that can be done for both least squares loss and KL-divergence is normalizing the factors and absorbing the scaling into the solved factor before each AO iteration. This also makes the subproblem that is being solved strictly convex in the case of KL-divergence (probably well-known, but see the tensor case: Chi, Eric C., and Tamara G. Kolda. "On tensors, sparsity, and nonnegative factorizations." SIAM Journal on Matrix Analysis and Applications 33.4 (2012): 1272-1299 ). This might be something to incorporate before doing the experiment suggested above.

4.) This is another very important point in my opinion. The mSOM algorithm is motivated/derived by considering the quadratic loss function in equation (6). Note that in equation (5) it was mentioned that we have a quadratic constraint, in eq(6) the constraint is implicitly assumed. To minimize (5) via a second order method (using Hessian), one may use a projected Newton method (Bertsekas, Dimitri P. "Projected Newton methods for optimization problems with simple constraints." SIAM Journal on control and Optimization 20.2 (1982): 221-246) for quadratic convergence. The algorithm would select active sets based on gradients and solve different sub matrices (of RxR gram matrix) for each column $h$ in parallel. This leads to a higher computational cost of  $O(NR^3)$ per iteration and thus one proposes to use these diagonal approximations (key point that should be mentioned in the paper which is somewhat vaguely out there. Precise cost would be good.). This Newton's iteration needs several iterations to converge unlike the alternating least squares without constraints which converges in one iteration due to objective being quadratic.

When we have a diagonal approximation with nonnegative entries, it allows for perfect decoupling of all the variables and we get efficient closed-form updates of the form given in equation (7). Note that if we don't merely try to decrease the loss function but minimize it for each AO subiteration, the algorithm needs several iterations (inside the AO), unlike performed in this paper. Chi and Kolda show that performing multiple of these is often better in terms of total number of AO iterations needed.

Also, it is nice that we can minimize the $1$-norm for majorants of the form $\text{Diag}(\mathbf{Bu}/\mathbf{u})$ as that leads to tight majorants of this form. But it is not clear why diagonal majorants of these forms are preferred. If we are looking for tightness one may want to minimize the largest or sum of eigenvalues of $\mathbf D - \mathbf B$ , where $\mathbf D$ is diagonal and $\mathbf D - \mathbf B$ is SPD.
Also, note that these are locally tight, and as I mentioned we need several Newton's iterations even in the quadratic case as it is not easy to figure the active set to minimize in one shot, so it is somewhat unclear if this tightness would always lead to better convergence (not mentioned in the paper).


5.) As I mentioned above that the main motivation for these approximations is computational efficiency, please provide a computational complexity analysis of the method (would be good to include sparse matrices case). Even though it is trivial / similar to MU, it is a good idea to have it in the paper.

---

> ### Author Response · Authors · 2026-01-29
> **Answers to reviewer 3f7V**
>
> **1.** *This is one of the most important points, please make the plots
> larger, with legends clearer (use more differentiable
> linestyle/markers). Also, it would be good to see the variance of
> different algorithms. The current plots only include medians.*
>
> **Answer:** We agree that the plots are overall hard to read. We have
> made them bigger, and revamped the layout so that they are also clearer.
> We have removed the linestyles for now, as for some reason the
> interpolation on the time grid interfers with the log scale on the x
> axis and the linestyles and the rendering is ugly. We might have time to
> fix this in the coming weeks, but we focused on the other issues for
> now.
>
> In our experience, showing error bars, and in fact showing several
> convergence plots, is not straightforward. In the simulated experiments,
> the algorithms are used to solved various problems that may have
> slightly different optimal loss. The non-convexity of the problem for
> NMF also means different local minima are typically reached, even for
> the same algorithm, if initialized several times. Therefore, there are
> cases where this problem/initialization variability is larger than the
> different between algorithms; even in cases where algorithm A may
> consistently outperform algorithm B, their error bars across several
> problems may overlap significantly, blurring the conclusion.
>
> We tried to address this issue by showing \"median\" plots, that are,
> for each iteration/time (on an interpolation grid defined per
> algorithm), the median loss value reached by each method. We agree
> nevertheless that this view only provides partial information. We
> observed that, in most of our experiments, error bars were reasonably
> small, or did not hinder the lisibility of the plots. Therefore we added
> $25\%$ and $75\%$ quantiles around the median plots for the synthetic
> experiments. For the experiments on actual dataset, we rather show the
> minimum error across time, since \"running the experiment ten times and
> picking the best run\" is a typical procedure for using NMF in practice.
> The error bars here show the difference between this minimum error and
> the median error.
>
> Convergence plots also tell a partial story, even with error bars. We are not showing an actual algorithm run (showing the median of a whole
> curve does not make sense). A different representation of the results
> that we wanted to explore is customized performance plots. We count how
> many times each algorithm is the fastest to reach a given threshold
> (usual performance plots would measure the number of time each methods
> has reached $x\%$ of the best method at the last iteration only). The
> threshold are determined based on the range of cost function values
> reached by the algorithms across iterations. These plots do not tell by
> how much an algorithm won versus the others, but can help appreciate in
> which scenario an algorithm is overwhelmingly better, and in which
> scenario there is no clear winner, the latter being hard to see on
> convergence plots. We have added these performance plots in the
> manuscript when they help with the interpretation of the results.

---

> ### Author Response · Authors · 2026-01-29
> **Answers to reviewer 3f7V (continued)**
>
> **2.** *The mSOM algorithm appears to be doing well for small noise
> (almost exact matrix-factorization), but is not too different from
> others when noise is added (which is more practical). I propose a
> synthetic experiment that might provide some insight in what cases mSOM
> might be better than MU: Since the majorization focuses on minimizing
> the $1$-norm of the diagonal matrices, it would be good to see what
> happens if the constructed synthetic matrices have differences in scales
> (happens in practical scenarios), i.e., scale the columns of the factors
> by large amounts making the underlying Hessian somewhat ill-conditioned.
> It is likely that mSOM does better than MU in this case, with some
> intuition why that might be the case.*
>
> **Answer:** Intuitively, mSOM and MU are different when $u=h$ (MU) is
> different from $u=1$ (mSOM) in the Hessian upper-approximation. This
> happens in particular when the components have different scales, but it
> also happens more generally for sparse components or structured problems
> such as encountered in HSI and audio where matrix $H$ contains
> respectively images and time activations. Therefore, the unbalancing
> experiment may not necessarily provide significantly different results
> from the other experiments, and in particular, AmSOM may not benefit
> from unbalancing. For the KL-divergence, the unbalanced experiment
> (performed by scaling the columns of matrix $W$ with exponentially
> decreasing weights) did not favor AmSOM over AMU, while the general
> conclusion is unchanged. For dense data, we observed that the balanced
> case already favors mSOM, and adding exponentially decreasing weights on
> the components did not change our conclusions, although the advantage of
> mSOM is slightly more pronounced.
>
> We also conducted the same experiment for the Frobenius loss, both with
> sparse and dense factors. The conclusions are similar: scaling the
> columns of matrix $W$ has an effect on the convergence plots, but the
> best methods remain AHALS and ANeNMF, with AmSOM and APGD being close
> competitors.
>
> The tests are available in the manuscript, in Appendix E.
>
> Also note that when noise is added, mSOM still outperforms MU in the
> dense case, it is simply less visible. Performance plots that have been
> added help to visualize this fact. In contrast, AMU struggles to beat
> AMUSOM, which indicates that in noisy scenario the search direction used
> by mSOM may not be better than the search direction of AMU.
>
> **3.** *This is in relation to the above experiment. Note that NMF is
> non-unique, i.e., I can absorb diagonal scaling and its inverse to both
> the factor rows without changing the factorization. When alternating
> optimization is performed, it choses 'an' element of the equivalence
> class of all the non-unique factorizations and optimizes things. This
> may work ok in practice, but might or might not get into trouble for
> ill-conditioned Hessians. One nice thing that can be done for both least
> squares loss and KL-divergence is normalizing the factors and absorbing
> the scaling into the solved factor before each AO iteration. This also
> makes the subproblem that is being solved strictly convex in the case of
> KL-divergence (probably well-known, but see the tensor case: Chi, Eric
> C., and Tamara G. Kolda.. This might be something to incorporate before doing
> the experiment suggested above.*
>
> **Answer:** Thank you for this insightful remark. We agree that scale
> ambiguity is intrinsic to NMF and that normalizing one factor and
> absorbing the scaling into the other is a well-known and often useful
> strategy in alternating optimization schemes. This normalization step is
> indeed commonly used in multiplicative updates and related AO methods to
> improve numerical stability and conditioning.
>
> We would like to clarify, however, that while such normalization can improve conditioning in practice, it does not in general render the
> subproblem strictly convex. More precisely, if one considers the
> subproblem of estimating ${\bf H}$ from fixed ${\bf V}$ and ${\bf W}$
> under the constraints ${\bf H}\ge 0$ and $\|{\bf h}_n\|_2 = 1$ for all
> rows ${\bf h}_n$ (as in: Chi, Eric C., and Tamara G. Kolda. \"On
> tensors, sparsity, and nonnegative factorizations.\" SIAM Journal on
> Matrix Analysis and Applications 33.4 (2012): 1272-1299), the feasible
> set is non-convex due to the unit-norm equality constraints.
> Consequently, strict convexity does not hold in general, even though the
> objective function (e.g., least squares or KL-divergence) is convex in
> ${\bf H}$ without these constraints.
>
> We agree with the reviewer that normalization can be beneficial in practice and we have clarified this point in the revised manuscript. As this technique is closely aligned with the discussion in Section 4.3 regarding the initialization for the KL divergence, we have added a sentence in that section to acknowledge this approach. However, we would
> prefer not to incorporate it as an explicit step of the algorithm.

---

> ### Author Response · Authors · 2026-01-29
> **Answers to reviewer 3f7V (continued)**
>
> **4.** *This is another very important point in my opinion. The mSOM
> algorithm is motivated/derived by considering the quadratic loss
> function in equation (6). (...) This Newton's iteration needs several iterations to
> converge unlike the alternating least squares without constraints which
> converges in one iteration due to objective being quadratic.*
>
> **Answer:** This is overall a very nice remark. We clarified the
> relationship between diagonal Hessian approximation and NNLS in the
> text. There are many algorithms for computing NNLS, and in fact this
> paper introduces yet another one, mSOM. The workhorse algorithm for
> NNLS, the active-set proposed by Lawson and Hanson, is rather cheap per
> iteration (about the same complexity as mSOM) but can either be
> extremely efficient if the initial support is close to the optimal one,
> or extremely slow if all the possible supports are explored (which may
> happen, the number of iterations is therefore non-polynomial in the
> problem size). We compare our method with HALS later in the manuscript,
> which is a state-of-the-art NNLS solver for NMF problems.
>
> *Moreover, we are working on making use of nonnegative least squares to
> solve KL-divergence problems with iterative second-order methods. While
> no existing work is doing exactly that to the best of our knowledge,
> Virtanen an co-authors have used a somewhat similar idea in the NMF
> audio community [@virtanenActiveSetNewtonAlgorithm2013].*
>
> When we have a diagonal approximation with nonnegative entries, it
> allows for perfect decoupling of all the variables and we get efficient
> closed-form updates of the form given in equation (7). Note that if we
> don't merely try to decrease the loss function but minimize it for each
> AO subiteration, the algorithm needs several iterations (inside the AO),
> unlike performed in this paper. Chi and Kolda show that performing
> multiple of these is often better in terms of total number of AO
> iterations needed.
>
> **Answer:** *We do perform several iterations of each block update. In
> fact an earlier version of this manuscrit discussed the number of inner
> iterations to perform for optimal speed, but the discussion was lengthy
> and we settled for using, in general, ten inner iterations. This can be
> reduced if the size of the dataset is not large.
>
> Also, it is nice that we can minimize the 1-norm for majorants of the
> form $\text{Diag}(Bu/u)$ as that leads to tight majorants of this form.
> But it is not clear why diagonal majorants of these forms are preferred.
> If we are looking for tightness one may want to minimize the largest or
> sum of eigenvalues of $D-B$, where $D$ is diagonal and $D-B$ is SPD.*
>
> **Answer:** We agree that this choice is somewhat arbitrary, although
> other choices within the family that optimize metrics on $\frac{Bu}{u}$
> are not straightfoward to derive, see also the answer to reviewer 2 and
> 3 first questions. Regarding the suggestion to minimize some metric over
> $D-B$, it would indeed be very interesting to do so if it can be done
> efficiently, as it yields a more direct formulation of diagonal
> approximation to the Hessian matrix. It is unclear to us, at the moment,
> how to solve this semi-definite program efficiently, or if this research
> idea has already been pursued in the context of NMF. This could be an
> interesting idea to pursue for later research.
>
> *Also, note that these are locally tight, and as I mentioned we need
> several Newton's iterations even in the quadratic case as it is not easy
> to figure the active set to minimize in one shot, so it is somewhat
> unclear if this tightness would always lead to better convergence (not
> mentioned in the paper).*
>
> **Answer:** As mentioned above we use several iterations of mSOM for
> each matrix $W$ and $H$ before changing blocks, although we are not
> technically performing AO. AmSOM can be understood as an AO algorithm
> where mSOM is used to solve each regression problem alternatively. This
> is why we introduce mSOM first in the manuscript, and performs
> experiments with NLS aside NMF. We expect that a better approximation of
> the cost function (tightness of the local approximation) leads to fast
> updates within each block update. This is shown theoretically by the
> linear convergence rate, and numerically with the NLS problems.
>
> Wether AO (solving each block update exactly) should be preferred to BCD
> (only reduce the cost for each block sequentially) for computing NMF is
> another interesting topic, that we do not address in this work.
>
> The questions raised by the reviewer are of interest to all
> readers. We added content in the manuscript to clarify these various
> points.

---

> ### Author Response · Authors · 2026-01-29
> **Answers to reviewer 3f7V**
>
> **5.** *As I mentioned above that the main motivation for these
> approximations is computational efficiency, please provide a
> computational complexity analysis of the method*
>
> **Answer:** The computational complexity of mSOM to update a vector
> ${\bf h}\in\mathbb{R}_+^{R}$ for a data vector ${\bf v}\in\mathbb{R}^M$
> is derived as follows. We assume that the matrix-vector product
> ${\bf W}^\top{\bf h}$ has a computational complexity of
> $\mathcal{O}(MR)$ by counting the number of products, although whether
> this is accurate for modern computer architecture is debatable. The mSOM
> algorithm relies on the following expensive computations:
>
> -   ${\bf W}{\bf{h}}$: the complexity is $\mathcal{O}(MR)$.
> -   ${\bf W}{\bf W}^\top{\bf h}$ and
>     ${\bf W}{\bf W}^\top{1}_R$: unless dimension $R$ is large
>     ($R$ in general is reasonably small for NMF problems), one may
>     precompute ${\bf W}{\bf W}^\top$ since this same quantity is used
>     across all mSOM iterations. This costs $\mathcal{O}(MR^2)$. The
>     product with vectors ${\bf h}$ or ${1}_R$ requires
>     $\mathcal{O}(R^2)$ operations.
>
> Therefore, mSOM for Frobenius loss requires initially
> $\mathcal{O}(MR^2)$ products. For a total of $k$ iterations, mSOM
> requires additionnaly $\mathcal{O}(k(R^2+R))$ products. The total
> complexity is therefore $\mathcal{O}((M+k)R^2)$. This is exactly the
> same assymptotic complexity as the MU algorithm, although mSOM involves
> an additional substraction. Therefore mSOM is bound to be slightly
> slower than MU.
>
> The costly operations in mSOM for the KL-divergence are
>
> -   ${\bf W}^\top{\bf h}$ that appears the numerator and denominator,
>     with complexity $\mathcal{O}(MR)$,
> -   ${\bf W}\odot {1}_{R,R}{\bf W}$, that costs
>     $\mathcal{O}(MR)$,
> -   ${\bf W}(\frac{{\bf v}}{{\bf W}^\top{\bf h}}-{1}_R)$ that
>     costs $\mathcal{O}(MR)$,
> -   $\left({\bf W}\odot {1}_{R,R}{\bf W}\right) \frac{{\bf v}}{\left({\bf W}^\top{\bf h}\right)^2}$
>     that also costs $\mathcal{O}(MR)$.
>
> Unlike in the Frobenius case, there is no simple way to precompute
> costly operations. When running mSOM for $k$ iterations, the complexity
> of mSOM is therefore $\mathcal{O}(kmr)$. The mSOM algorithm in the
> KL-divergence case is therefore typically slower than in the quadratic
> case when $R\ll k$ and $R \ll M$. The MU algorithm for KL-divergence
> requires only two operations in $\mathcal{O}(mr)$, namely products
> ${\bf W}^\top{\bf h}$ and ${\bf W}\frac{{\bf v}}{{\bf W}^\top{\bf h}}$,
> and is therefore slightly faster per iterations.
>
> This discussion has been added to the manuscript.
>
> ## Additional Comments
>
> Small corrections below
> -   Proposition 1, no need to say $B$ has nonnegative entries (already
>     said).
> -   Proof of Prop 1: replace x by h
> -   brackets appear as \]\[ instead of \[\]
>
> **Answer:** Thank you for spotting these typos; they have been
> corrected.

---

> ### Comment · Reviewer_3f7V · 2026-02-11
> **Regarding convexity of the objective**
>
> Thanks for responding to my comments. Maybe one of my comment regarding convexity was unclear. I meant to say when you are solving for one of the factors while the other is normalized, you do not need the unit norm constraint. Let $\mathbf{\bar{W}}$, and $\mathbf{\bar{H}}$ be the factors with unit norm columns and the norms be absorbed into $\mathbf{\Lambda}$, a diagonal matrix, then when we solve for $\mathbf{W} = \mathbf{\bar{W}} \mathbf \Lambda$, we wont need the unit norm constraint in the solve. The Hessian of the minimization problem will be a semi positive definite matrix, and adding a regularization should lead to a strictly positive definite matrix, i.e., a strictly convex subproblem. See section 3.3 of Chi, Eric C., and Tamara G. Kolda. "On tensors, sparsity, and nonnegative factorizations." SIAM Journal on Matrix Analysis and Applications 33.4 (2012): 1272-1299)
>
> I hope it is clear that the minimization problem (with nonnegativity) that we solve for is not constrained with unit norm columns, but the scaling is automatically handled in the alternating optimization regime when we normalize the columns of the factor which we are not solving for in that subiteration.

---

> > ### Author Response · Authors · 2026-02-13
> >
> > Thank you for clarifying the comment. We understand that, e.g., in the factorization $WH^T$, when matrix $H$ is updated inside an alternating algorithm, the norms of columns of $W$ can be pulled in $H$ implicitly by algorithmic normalization of matrix $W$. As long as columns of matrix $W$ are nonzero, the subproblem of estimating $H$ is always strictly convex (not strongly convex since the KL-divergence has curvature that goes to zero at infinity). When a component is zeroed out, however, strict convexity does not hold, and normalization is ill-defined.
> >
> > In our setup, the $\epsilon$ constraint prevents zero components, so this algorithmic normalization is always valid, and the subproblems are always strictly convex. Algorithmic normalization, therefore, does not change the strict convexity of the subproblems. We do not require normalization of the columns of $W$ in the proofs of convergence of mSOM, but it could be a useful hypothesis if a dedicated proof of convergence for AmSOM were to be performed. The work of Chi and Kolda is concerned with the convergence of an algorithm solving the problem with respect to all blocks, not a single subproblem, and, as far as we understand, makes use of normalization (in the sense of pulling norms into the updated matrix) in this context. Their problem setup is also different. The column normalization constraint is important to interpret the scaling parameters $\lambda$ as weights, whereas this is not the case in our work.
> >
> > We do not expect this normalization to also change the geometry of the problem; the updates obtained by mSOM should be the same (we have not checked this formally). One reason why one would perform normalization in our setup would be for numerical stability. While we did not experiment with this idea on the practical side, one of the authors has extensive experience with nonnegative tensor factorizations, has tried this approach several times in popular algorithms, and has not observed convincing differences compared to not performing such a normalization.

---

### Review · Reviewer_SxTQ · 2025-12-22

**Summary Of Contributions:**

The paper describes an algorithm to train non-negative matrix factorization, a classical topic used in several area like early topic models. The paper inherits alternating minimization and a seem to me a majorization-minimization paradigm? ( not sure this is the term). The key operation is to choose the eigen-vectors to characterize local curvature that uppers bounds the objective function. The experiments showed good improvement over prior methods.

**Audience:**

Yes

**Audience Explanation:**

NMF is a classical machine learning tasks that applies to wide range of tasks, and fundamental different from PCA/sparse coding, which leads to specialized optimizers for this area.

**Broader Impact Concerns:**

non

**Claims And Evidence:**

Yes

**Claims Explanation:**

The paper discussed prior methods, why they choose this option, the convergence rate is proven. experiments are small, but acceptable.

**Requested Changes:**

sorry for a late review. I tried to read it several times, each time I was blocked by parts difficult to understand. So please correct me if I am wrong.

The paper describes an algorithm to train non-negative matrix factorization, a classical topic used in several area like early topic models. The paper inherits alternating minimization and a seem to me a majorization-minimization paradigm? ( not sure this is the term). The key operation is to choose the eigen-vectors to characterize local curvature that uppers bounds the objective function.

I don't quite understand the naratives that leads to the main algorithm, in EQ.9, why is this an minimizer of 'median' by minimizing over L1?

Is there a theoretical advantage of this algorithm over 1, u(h)=h, 2, u(h) largest singular value, on top of the linear convergence rate that is proven here, like, e.g. a better convergence rate constant?

Is there a geometric interpretation why is design choice is better?

I would say the experiments are a little bit of small in scale considering several finished within 2 seconds, but I won't be picky in this.

---

> ### Author Response · Authors · 2026-01-29
> **Answers to reviewer SxTQ**
>
> **Answer:** We apologize if some parts were difficult to follow. We have
> revised the text to improve clarity, and hope the presentation is now
> clearer.
>
> **1.** The paper describes an algorithm to train non-negative matrix
> factorization, a classical topic used in several area like early topic
> models. The paper inherits alternating minimization and a seem to me a
> majorization-minimization paradigm? ( not sure this is the term). The
> key operation is to choose the eigen-vectors to characterize local
> curvature that uppers bounds the objective function.
>
> I don't quite understand the narratives that leads to the main
> algorithm, in EQ.9, why is this an minimizer of 'median' by minimizing
> over L1?
>
> **Answer:** The goal of the proposed second-order type algorithm is to
> accelerate convergence while maintaining low computational complexity.
> In this sense, the proposed method indeed follows a MM strategy,
> although with a specific focus on constructing efficient diagonal
> second-order surrogates.
>
> More precisely, Proposition 1 shows that a family of diagonal majorants
> of the Hessian of the objective function can be constructed in the form
> ${\operatorname{Diag}}\left(\frac{{\bf B}{\bf u}}{{\bf u}}\right)$,
> for ${\bf u}>0$, which upper-bounds the local curvature of the loss.
> This result also explains why multiplicative updates naturally arise in
> this framework, as they correspond to particular choices of the vector
> ${\bf u}$.
>
> Motivated by this observation, our objective is to select a strictly
> positive vector ${\bf u}$ such that the resulting diagonal majorant is
> as close as possible to the true Hessian, while remaining
> computationally efficient. A natural first idea would be to minimize the
> $\ell_2^2$ distance between them. This sub-optimization problem does not have a closed-form solution.
> Moreover, we can see that it forces the diagonal majorant to the
> diagonal of the true Hessian. We then decided to use $\ell_1$-norm
> instead of $\ell_2^2$. This has been expanded in the main text.
>
> For this choice of $u$ we can have the closed-form of optimizer vector,
> avoiding any additional inner optimization loop. Moreover, we think this
> can also be good when the distribution of value in the Hessian matrix is
> sparse. To illustrate the behavior of these two strategies, we have
> prepared a supplementary document that provides illustrative figures to
> address this point. This document is provided alongside the revised
> manuscript, and the above derivations are now mentioned in the main
> document.
>
> An alternative choice for ${\bf u}$ is discussed in Appendix B,
> illustrating different trade-offs between approximation accuracy and
> computational cost.
>
> We agree that the choice of the optimal diagonal majorant is not unique,
> and that exploring alternative constructions could further improve the
> method. We therefore view this aspect as an interesting direction for
> future work.
>
> **2.** Is there a theoretical advantage of this algorithm over 1,
> u(h)=h, 2, u(h) largest singular value, on top of the linear convergence
> rate that is proven here, like, e.g. a better convergence rate constant?
>
> **Answer:** 1. $u(h)=h$ This choice corresponds to the classical
> multiplicative update (MU) rule. To the best of our knowledge, MU
> methods do not come with an explicit convergence rate guarantee, and
> their convergence analysis typically relies on monotonic decrease
> arguments rather than quantitative rates. As a result, no direct
> comparison in terms of convergence constants can be established with our
> approach.
>
> 2\. $u(h)$ is the largest singular value of the Hessian. This case is
> discussed in Appendix B.
>
> **3.** Is there a geometric interpretation why is design choice is
> better?
>
> **Answer:** This is an excellent suggestion, and we agree that a
> geometric interpretation could help clarify the choice of the $\ell_1$
> norm. However, given the current length and state of the manuscript, we
> decided not to include additional figures in the main manuscript. The
> answer to the first question in the supplementary material provides some
> additional intuition. The shape of the local majorants can be plotted
> for illustration, and we will use this idea for oral presentations.
>
> **4.** I would say the experiments are a little bit of small in scale
> considering several finished within 2 seconds, but I won't be picky in
> this.
>
> **Answer:** We agree that the experiments are relatively small in scale;
> however, they are designed specifically to analyze the convergence speed
> of the proposed method. At this scale, this objective can already be
> clearly observed. Moreover, the experiments on the audio and HSI data
> are medium-scale; running for instance mSOM with KL-divergence on the
> HSI dataset requires about one minute on a powerful laptop per
> experimental condition. Larger-scale experiments are certainly of
> interest and will be considered in future work, where regularization and
> prior information will also be taken into account to further improve the
> quality of the solutions.

---

### Review · Reviewer_sFAG · 2026-01-14

**Summary Of Contributions:**

The work introduces an elegant mathematical framework for NMF models under the principles of second-order majorant that accommodates both quadratic and beta-divergence losses. It constructs a tight quadratic upper bound for Frobenius loss function cases by majorizing the elementwise nonnegative Hessian matrix. Accompanying algorithms are alternating gradient descent ones similar to the celebrated AMU approach. Convergence analysis shows global convergence to stationary points for the alternating version of the algorithm. Some theoretical interpretation and practical algorithm version for beta divergence scenario are also discussed. Both synthetic and real data experiments are presented and compared with SOTA algorithms. Although the paper provides nice theoretical contributions to NMF literature, the empirical evidence is not convincing to advocate for practical use.

**Audience:**

Yes

**Audience Explanation:**

The paper makes a good attempt to contribute to the NMF literature which has many applications in several domains of machine learning/AI.

**Claims And Evidence:**

No

**Claims Explanation:**

The theoretical claims are more or less supported with convincing details. Whenever a design choice is made, the authors have taken efforts to justify the choice. Although theoretical claims are substantial, there are some gaps in convincing the readers about the empirical advantages of the approach. More questions/comments in this regard are explained in the "requested changes" tab

**Requested Changes:**

There are some key questions/comments which is important to be addressed.

1.	The authors choose median-based selection to choose the u vector which looks like having similar computational benefits when compared to MU. Why do you think this is computationally cheap? Why are the updates independent of current iterate h? Please clarify

2.	The work does not rigorously design an algorithm for beta divergence by falling back to standard MU to guarantee convergence. Although the authors acknowledge this, are there any potential directions to fix this gap which could be worthwhile to discuss to highlight the generality of this framework in beta divergence scenarios.

3.	There is discussion on the sparse datasets and how does mSOM algorithm supports this type of data in section 3.3. The discussion is more based on the beta divergence loss. How does the Frobenius loss-based mSOM supports sparse datasets? Are there any synthetic results with controlled sparsity to see how does these algorithms behave in sparse scenarios.

4.	The scaling laws proposed in Proposition 4 for beta divergence mSOM seems nice, but have not presented any experimental results to confirm its practicality. Have you checked this? If it has practical limitations, why is it so? In section 5.4.2, it is written “In the NMF problem, for the rank two case, the loss even increases. This can happen despite our efforts to initialize carefully the factor matrices.” What is the initialization used here?

5.	In the introduction, it is written “However, to the best our knowledge, mirror gradient descent
for NMF with ß-divergence loss does not outperform other baselines algorithms such as AMU”. Can you provide some empirical evidence for this? Is this statement general or particular to some specific data types? Please clarify.

6.	The experiments provides a quite mixed conclusions in terms of practical use where AMU and AHALS still stand as strong methods for NMF compared to the proposed method. Hence, what should be the final recommendations to the practioners if one were to choose mSOM-based algorithms?

7.	Can the algorithmic and theoretical framework of the proposed approach support any additional regularizations/prior knowledge on factor matrices other than nonnegativity? Please discuss some potential in this direction.

8.	Minor issues: There are some minor typos/definition missing/formatting issues: e.g.,

a.	“for many dataset” in Page 2

b.	Equation 6, it is better to define the last term

c.	“]0,2[“  in Page 6 and in several other places

d.	“argmin u>=0 ||” in Page 8 just above equation 15.

e.	The introduction section could be better reorganized with some paragraph headings such as prior work, contributions etc. In the current form, it is long and tedious.

---

> ### Author Response · Authors · 2026-01-29
> **Answers to reviewer sFAG**
>
> **1.** The authors choose median-based selection to choose the u vector
> which looks like having similar computational benefits when compared to
> MU. Why do you think this is computationally cheap? Why are the updates
> independent of current iterate h?
>
> **Answer:** We have extensively discussed this point in answer to
> reviewer 2's first question. To state things slighly differently, the
> choice of the \"median\" ${\bf u}$ is driven essentially by two factors:
>
> -   The vector ${\bf u}$ that leads to optimal quantity
>     $\|\frac{{\bf B}{\bf u}}{{\bf u}} \|_1$ is known in closed form and
>     the resulting Hessian approximation can be computed at about the
>     same computational cost as for the MU. Complexity has been added in
>     the manuscript; this point is now clearer.
>
> -   Other metrics for measuring the length of
>     $\frac{{\bf B}{\bf u}}{{\bf u}}$ either lead to harder problems (l2
>     loss) or fall back to well known algorithms. Indeed, $\ell_\infty$
>     gives gradient descent with Lipchitz-constant step-size in the
>     Frobenius norm case. In the KL-divergence case, it computes a local
>     Lipschitz constant. This scheme is reminiscent of adaptive gradient
>     descent [@malitskyAdaptiveGradientDescent2020b], but the computation
>     of the local Lipschitz constant at each iteration is costly. This
>     has also been clarified in the manuscript.
>
> The choice of ${\bf u}=1$ is indeed independent of the current iterate
> ${\bf h}$, which is surprising at first. However, information about the
> curvature of the current iterate ${\bf h}$ is contained in the Hessian,
> which mSOM approximates by computing the unweighted sum (${\bf u}=1$) of
> its rows, stored in a diagonal matrix. Our contribution is to show that
> this operation leads to a majoration of the Hessian, that convergence of
> the iterates can be guaranteed with a stepsize $\gamma\geq 1$, and that
> the weighted sum of the rows of the Hessian (${\bf u}={\bf h}$, *i.e*.
> MU) often leads to worse performance.
>
> **2.** The work does not rigorously design an algorithm for beta
> divergence by falling back to standard MU to guarantee convergence.
> Although the authors acknowledge this, are there any potential
> directions to fix this gap which could be worthwhile to discuss to
> highlight the generality of this framework in beta divergence scenarios.
>
> **Answer:** As we discussed in the manuscript, Theorem 2 establishes a
> local linear convergence result for the proposed mSOM algorithm. This
> convergence guarantee mainly depends on the initialization of each
> alternating optimization (AO) step. In practice, divergence from the
> admissible search domain is often caused by scaling issues, which are
> commonly handled using MU rules in the context of $\beta$-divergence.
>
> This type of safeguard is not uncommon in NMF Algorithm. For instance,
> similar strategies are employed in the work of Le Thi Khanh Hien and
> Gillis [@hien2021algorithms]. One could also adopt the type of
> stabilization strategy suggested in Point 3 of Reviewer 1.
>
> However, as discussed in our response to that point, although these
> techniques are effective in practice, they do not guarantee strict
> convexity of the resulting subproblems. Consequently, global convergence
> of the associated updates cannot be theoretically ensured.
>
> In practice, we observe that the proposed mSOM updates already provide
> sufficient control of the optimization process without requiring a
> fallback to MU. For this reason, we chose to retain this formulation in
> the present work. Nevertheless, incorporating scaling or safeguarding
> strategies, such as those proposed by Chi and Kolda (2012), represents a
> natural and promising direction to extend the framework and better
> support $\beta$-divergence formulations, and will be considered in
> future work.
>
> **3.** There is discussion on the sparse datasets and how does mSOM
> algorithm supports this type of data in section 3.3. The discussion is
> more based on the beta divergence loss. How does the Frobenius
> loss-based mSOM supports sparse datasets? Are there any synthetic
> results with controlled sparsity to see how does these algorithms behave
> in sparse scenarios.
>
> **Answer:** Sparsity in the data is typically an issue for
> beta-divergences with $\beta<2$ that are not gradient Lipschitz at zero.
> Sparse factors are also a difficulty for AMU that cannot compute zeros
> exactly. Running as experiment for fitting sparse factors with the
> Frobenius loss is therefore maybe less critical. Algorithms such as
> AmSOM, AHALS, APGD and ANeNMF can converge to solutions with zero values
> without issue.
>
> Nevertheless we ran the experiment for NMF with both sparse and/or
> unbalanced factors with Frobenius loss. The results are presented in the
> appendix of the manuscript, and essentially show that both AMU and
> AMUSOM perform very poorly in this case, while AmSOM has no particular
> issue and is competitive with other state-of-the-art algorithms. The
> choice of the proposed Hessian majorant is thus critical for Frobenius
> loss with sparsity.

---

> ### Author Response · Authors · 2026-01-29
> **Answers to reviewer sFAG (continued)**
>
> **4.** The scaling laws proposed in Proposition 4 for beta divergence
> mSOM seems nice, but have not presented any experimental results to
> confirm its practicality. Have you checked this? If it has practical
> limitations, why is it so? In section 5.4.2, it is written "In the NMF
> problem, for the rank two case, the loss even increases. This can happen
> despite our efforts to initialize carefully the factor matrices." What
> is the initialization used here?
>
> **Answer:** We have not checked numerically the scaling low of
> Proposition 4. The bound for the bassin of attraction is derived after
> several crude bounds, and is probably quite pessimistic. The practical
> consequence of the result, however, that linear convergence occurs only
> locally, is that the cost sometimes increases and that convergence is
> not fast when far from the solution. These two cases are illustrated in
> the experiments with the rank two case for the audio + NMF KL-divergence
> experiment and the rest of the NMF audio experiment, respectively. We
> have clarified that these experimental results are in line with the
> convergence theory we developed. Note that for the rank-two case,
> initialization is performed in the same way as other real data
> experiments: a hand-picked matrix $W$ is chosen from pretrained
> templates, and noise is added. For rank two, this initialization might
> not be very good, as choosing two templates to represent a whole song is
> unrealistic, and we did not optimize over which templates are chosen
> (here we chose MIDI notes 27 and 28, that are quite low-pitch). This has
> also been clarified in the manuscript.
>
> Studying empirically the bassin of attraction is typically difficult in
> high dimensions, and very problem-dependent. We would not perform such
> an experiment with enthousiasm.
>
> **5.** In the introduction, it is written "However, to the best our
> knowledge, mirror gradient descent for NMF with ß-divergence loss does
> not outperform other baselines algorithms such as AMU". Can you provide
> some empirical evidence for this? Is this statement general or
> particular to some specific data types? Please clarify.
>
> **Answer:** Mirror descent for minimizing KL-divergence is interesting
> because while KL-divergence is not Lipschitz smooth (in particular at
> zero), it is smooth relative to Burg entropy. In the original work of
> Bauschke, Bolte and Teboulle [@bauschkeDescentLemmaLipschitz2017], this
> is introduced without any numerical simulation. Some works have later
> used Burg entropy to perform mirror descent with KL-divergence in the
> context of data-driven regularizations, as it allows to derive
> convergence results for first-order methods with proximal
> regularizations [@huraultConvergentBregmanPlugandPlay2023b]. In our
> experience, backed solely by various discussion within the community,
> Burg entropy leads to a loose majorization of the cost function. Without
> further algorithmic tricks such as extrapolation, mirror descent is
> consistently slower than MU. Regarding NMF and mirror descent, Takahashi
> and co-authors have used Burg entropy with additional quadratic
> regularization, but to design a joint majorisation of the cost with
> respect to both matrices $W$ and $H$. Their approach is therefore not a
> block-coordinate algorithm that we restrict to in this manuscript;
> moreover their algorithm does require extrapolation to, sometimes,
> outperform (unextrapolated) MU. Another recent work leverages mirror
> descent to provide convergence proofs for extrapolated
> algorithms [@hienBlockMajorizationMinimization2025].
>
> Our conclusion is that mirror descent is a promising research direction
> to build convergent majorization-minimization algorithms for NMF. Still,
> there is no proof in the literature that mirror descent with Burg
> entropy. We have clarified our position and added relevant references in
> the manuscript. Clearly there is a lack a published comparison including
> Burg entropy mirror descent, therefore we were careful with our wording.
>
> **6.** The experiments provides a quite mixed conclusions in terms of
> practical use where AMU and AHALS still stand as strong methods for NMF
> compared to the proposed method. Hence, what should be the final
> recommendations to the practioners if one were to choose mSOM-based
> algorithms?
>
> **Answer:** Several experiments have been added, and the conclusions of
> the experiments are now clearly stated in the manuscript. See the
> general answer above for more details.

---

> ### Author Response · Authors · 2026-01-29
> **Answers to reviewer sFAG (end)**
>
> **7.** Can the algorithmic and theoretical framework of the proposed
> approach support any additional regularizations/prior knowledge on
> factor matrices other than nonnegativity? Please discuss some potential
> in this direction.
>
> **Answer:** Yes, the proposed framework can in principle support
> additional regularization or prior knowledge on the factor matrices.
>
> In this work, however, we deliberately focus on a formulation involving
> only the fidelity term, in order to clearly analyze the behavior and
> efficiency of the proposed majorant. In this setting, the main
> contribution lies in the use of a diagonal Hessian approximation, which
> leads to a particularly efficient and stable inversion step.
>
> In particular, the diagonal approximation is well-suited for extensions
> involving separable regularization terms. Indeed, it naturally enables
> the use of proximal or projection updates within a forward-backward or
> proximal-gradient framework, making it possible to incorporate various
> priors such as sparsity penalties or norm-based constraints.
>
> While such extensions are not explored in the present paper, the
> simplicity and efficiency of the diagonal approximation make it
> especially attractive for integrating additional regularization terms in
> future work.
>
> ## Minor issues
>
> There are some minor typos/definition missing/formatting issues: e.g.,
>
> -   "for many dataset" in Page 2
>
> -   Equation 6, it is better to define the last term
>
> -   "\]0,2\[" in Page 6 and in several other places
>
> -   "argmin u\>=0 \|\|" in Page 8 just above equation 15.
>
> -   The introduction section could be better reorganized with some
>     paragraph headings such as prior work, contributions etc. In the
>     current form, it is long and tedious.
>
> **Answer:** The typos have been corrected. A subsection heading
> \"Contributions\" has been added to structure the introduction,
> hopefully the introduction is now slightly more digest. We could further
> move some references in a dedicated related works paragraph, with
> substantial rewriting. We believe however that in its current form, the
> context is clear the contributions are properly introduced with respect
> to the state-of-the-art.

---

### Author Response · Authors · 2026-01-29
**Response to reviews**

# General answer

We thank the reviewers for their careful reading of the
manuscript and their insightful comments that helped improve the quality
of the manuscript and the credibility of the numerical experiments. Here
is a list of modifications on the manuscript:
-   Experiments:
    -   Adding clarifications on the experimental claims: mSOM is
        competitive with the state-of-the-art for Frobenius norm (both
        sparse and dense), performs favorably wrt MU for KL in the
        synthetic experiments, but struggles with realitic dataset (the
        audio experiment still to be further understood when it fails in
        that setting; this application is reported as special for
        algorithmic performance also by e.g. Hien and
        Gillis [@hien2021algorithms]).
    -   Adding experiments to support these claims: 1. HSI with the
        KL-divergence, which a dense, large-scale experiment for KL 2.
        Audio with the Frobenius norm, which is a sparse, medium-scale
        experiment for the Frobenius norm. 3. Sparse + unbalanced
        Frobenius loss simulated experiments. These experiments complete
        the simulations already provided and hopefully better highlight
        the strengths of the mSOM algorithm.
    -   Additional information: winner plots (customized performance
        plots) and error bands on convergence plots.
    -   The plots are now larger; they can be further improved (although
        this is quite time-consuming) before sending the final
        manuscript if required.
-   The complexity of mSOM updates is now explicitly written in the
    manuscript.
-   The link between NNLS solutions, separable constraints and diagonal
    Hessian approximations is now more detailed.
-   Clarifications around the choice of the Hessian majorant. In
    particular, we identify that mSOM approximates explicitly the
    Hessian with the $\ell_1$ norm.
-   More context around the use of mirror descent for NMF with
    $\beta$-divergences.
-   The fact that AmSOM is an approximate Alternating Optimization
    algorithm is now more clearly stated.
-   Various typos were corrected.
-   The introduction is now structured with a contributions section
-   A sentence about normalization of the columns/rows has been added to
    acknowledge this option.

Finally, we are open to trying the method on more dataset, but plan to
do so on a larger scale within the benchopt initiative, comparing also
more NMF algorithms that are presented in this manuscript. We hope the
present set of experiments is sufficient to convince the reader to try
our method, despite the lack of convincing results for AmSOM with
$\beta$-divergences on realistic dataset.


More detailed answers to each review are provided in responses below.

## References
Heinz H. Bauschke, Jérôme Bolte, and Marc Teboulle. A Descent Lemma Beyond Lipschitz Gradient Continuity: First-Order Methods Revisited and Applications. Mathematics of Operations Research, 42(2):330–348, May 2017.

Le Thi Khanh Hien and Nicolas Gillis. Algorithms for nonnegative matrix factorization with the Kullback–Leibler divergence. Journal of Scientific Computing, 87(3):1–32, 2021.

Le Thi Khanh Hien, Valentin Leplat, and Nicolas Gillis. Block Majorization Minimization with Extrapolation and Application to β-NMF. SIAM Journal on Mathematics of Data Science, pp. 1292–1314, September 2025. 11 Under review as submission to TMLR

Samuel Hurault, Ulugbek Kamilov, Arthur Leclaire, and Nicolas Papadakis. Convergent Bregman Plug-and-Play Image Restoration for Poisson Inverse Problems. Advances in Neural Information Processing
Systems, 36:27251–27280, December 2023.

Yura Malitsky and Konstantin Mishchenko. Adaptive Gradient Descent without Descent. In Proc. 37th Int. Conf. Machine Learning. PMLR, 2020.

Tuomas Virtanen, Jort Florent Gemmeke, and Bhiksha Raj. Active-Set Newton Algorithm for Overcomplete Non-Negative Representations of Audio. IEEE Transactions on Audio, Speech, and Language Processing, 21(11):2277–2289, November 2013.

---

### Decision · Action_Editor_QHwZ · 2026-04-07

**Recommendation:** Accept as is

**Audience:**

Yes

**Audience Explanation:**

There is still enough interest in NMF.

**Claims And Evidence:**

Yes

**Claims Explanation:**

An NMF algorithm based on second order majorant is proposed. Theoretical claims are solid, and experimental results are ok, although not very impressive.